# PSC: Efficient Grammar-Constrained Decoding via Parser Stack Classification

## Abstract

LLMs are widely used to generate structured output like source code or JSON. Grammar-constrained decoding (GCD) can guarantee the syntactic validity of the generated output, by masking out tokens that violate rules specified by a context-free grammar. However, the online computational overhead of existing GCD methods, with latency typically scaling linearly with vocabulary size, limits the throughput of LLMs, especially for models with large vocabularies. To address this issue, we propose PSC, a novel grammar-constrained decoding method. By combining acceptance conditions of all vocabulary tokens into a single classifier of the parser stack during preprocessing, PSC can compute the complete vocabulary mask by checking the parser stack exactly once per decoding step, with time complexity independent of the vocabulary size. Experiments show that PSC computes masks up to 770× faster than baselines on complex programming language grammars, and up to 30× faster for schema-conformant JSON; end-to-end LLM throughput with PSC approaches that of unconstrained decoding.

## 1 Introduction

In recent years, the ability for Large Language Models (LLMs) to generate structured output has been widely recognized and utilized (Qwen et al.; Grattafiori et al.; Gemma Team et al.). Source code can be viewed as structured output that adheres to the syntax of programming languages, and LLM-based coding assistants, such as GitHub Copilot (GitHub) and Cursor (Anysphere Inc.), have been widely adopted by developers to assist in writing code to improve their productivity. When LLMs are used as a tool, users often expect the generated output to conform to a specific format, such as Markdown or JSON with custom schemas (Liu et al., 2024; vLLM Team; OpenAI). All of these applications rely on the ability of LLMs to generate output that adheres to a specific syntax.

However, generating in a structured format is complex, as it requires not only understanding the semantics of given input but also adhering to the specific grammars of target formats. Since language models are essentially probabilistic models, there is no guarantee that the generated output will always conform to the required grammar.

To address this issue, *grammar-constrained decoding* (GCD) (Geng et al., b; Scholak et al.; Poesia et al.; Ugare et al.) is proposed to ensure that the generated output always conforms to the specified context-free grammar. A GCD method works by incorporating a grammar checker into the decoding process, as shown in Figure 1a. At each decoding step, the checker determines which tokens in the vocabulary can be appended to the current prefix while not violating the grammar. The logits generated by the language model are then masked to only allow the valid tokens, and the next token is generated by sampling from the masked logits.

The overhead of GCD is determined by the newly introduced step of validity calculation. A naive implementation, as shown in Figure 1b would require calling the parser for *every* token in the vocabulary to check its validity, resulting in a time complexity of $\mathcal{O}(|\mathcal{V}|)$ per decoding step, where $|\mathcal{V}|$ is the vocabulary size. This can add significant overhead, especially for large vocabularies in modern language models, e.g. 128k tokens in Llama-3 (Grattafiori et al.), 151k tokens in Qwen series (Bai et al.), or 262k tokens in Gemma 3 (Gemma Team et al.). The overhead is particularly pronounced for smaller models, where the time taken by model inference is relatively small.

(a) Decoding step in grammar-constrained decoding (GCD).

(b) Naive GCD implementation.

(c) Our GCD method PSC.

Figure 1: An illustration of grammar-constrained decoding, showing (a) the overall working process, (b) the naive implementation that directly simulates the PDA, and (c) our method PSC that precomputes the DFA and the valid token masks.

To speed up GCD, various techniques have been proposed in the literature, summarized in Section 2.4. However, none of them can fundamentally change the $\mathcal{O}(|\mathcal{V}|)$ worst-case time complexity while maintaining correctness.

We propose a novel GCD method PSC that replaces the repetitive runtime parsing over the whole vocabulary with a one-time classification of the current parser stack, as shown in Figure 1c. The checking process of the parser can be seen as a function of both the token and the state of the parser, which is usually a stack. For each token, our method PSC constructs a finite-state automaton (FSA) that represents the exact requirements on the parser stack to accept that token, i.e., the FSA accepts a parser stack if and only if that token is accepted by a parser with that stack. All these FSAs can then be combined into a single FSA that classifies the parser stack into a finite number of classes, each corresponding to a different vocabulary mask. During decoding, we only need to check the parser stack exactly once per decoding step to get the vocabulary mask, which is ready to be applied to the logits. This eliminates the need to call the parser for each token in the vocabulary, resulting in a significant speedup.

We conduct extensive experiments on grammar-constrained decoding in Java, Go, SQL, and schema-conformant JSON to evaluate the efficiency of our method. Compared to the current state-of-the-art method LLGuidance, our method achieves up to 770 times speedup in mask computation on complex programming language grammars, and up to 30 times speedup for schema-conformant JSON generation. In the end-to-end decoding throughput experiments, the throughput of PSC approaches that of unconstrained decoding, and is significantly higher than LLGuidance, especially on smaller models and larger batch sizes.

In summary, our contributions are as follows:

- We propose a novel GCD method PSC that leverages finite-state automata to classify parser states to use the precomputed vocabulary mask, significantly reducing the time overhead of grammar-constrained decoding.

- We provide a theoretical analysis of PSC. We prove that the set of the parser stacks that can accept a given token can be formally described as a regular language. This justifies the correctness of our method and provides a theoretical foundation for future research on grammar-constrained decoding.

- We demonstrate the efficiency of PSC through extensive experiments on grammar-constrained decoding in Java, Go, Python, and schema-conformant JSON, achieving significant speedup in mask computation compared to existing techniques; end-to-end decoding throughput with PSC approaches that of unconstrained decoding.

## 2 BACKGROUND AND RELATED WORK

We introduce the task of grammar-constrained decoding in this section, give some brief and informal definitions of the concepts used in the paper, and then review the related work. **A quick lookup table for the symbols, notations, and the exact definitions can be found in Appendix A.2.**

### 2.1 THE TASK: GRAMMAR-CONSTRAINED DECODING

Let $\Sigma$ be the character set used by the language model, e.g. the Unicode. Given a prefix of tokens, the task of a language model is to generate the next-token distribution over the vocabulary $\mathcal{V} \subset \Sigma^+$. For a language $L \subseteq \Sigma^*$, the task of *constrained decoding* aims to generate a sample in $L$ from the language model. In each step, given prefix $x \in \Sigma^*$, it calculates the set of valid tokens in $\mathcal{V}$, i.e. tokens that, when concatenated after $x$, become a prefix of some strings in $L$.

$$c(x \in \Sigma^*) := \{v \in \mathcal{V} | \exists y \in \Sigma^*, xvy \in L\} . \tag{1}$$

When the language $L$ is defined by a context-free grammar, the task is called *grammar-constrained decoding* (Ugare et al.; Koo et al.; Park et al.; Moskal et al.). Determining whether a string is syntactically valid usually involves two phases: lexical analysis and syntax analysis[1] (Aho & Ullman). In lexical analysis, the lexer $\mathcal{T}$, usually modeled as a deterministic finite-state transducer (FST) (Aho & Ullman; Koo et al.; Park et al.), transduces the text $w \in \Sigma^*$ into a terminal sequence $\mathcal{T}(w) \in \Gamma^*$, where $\Gamma$ is the set of terminals. In syntax analysis, the parser $\mathcal{P}$, usually modeled as a terminating deterministic push-down automaton (PDA) (Aho & Ullman), determines whether a terminal sequence $x \in \Gamma^*$ is valid, here written as $x \in \mathcal{P}$. So we have

$$w \in L \iff \mathcal{T}(w) \in \mathcal{P}. \tag{2}$$

### 2.2 FINITE-STATE TRANSDUCER

The lexer $\mathcal{T}$ is a *finite-state transducer* (FST). It reads in the input string $w \in \Sigma^*$ character by character, and maintains a state $q \in Q$, where $Q$ is the finite set of states. When reading in the input character $c \in \Sigma$, the FST transits from state $q$ to state $q'$ and outputs a terminal sequence $t \in \Gamma^*$, written as $q \xrightarrow{c:t}_{\mathcal{T}} q'$. It may also transit without reading in any character, written as $q \xrightarrow{\varepsilon:t}_{\mathcal{T}} q'$.

### 2.3 PUSHDOWN AUTOMATA

The parser $\mathcal{P}$ is a *pushdown automaton*. It reads in the input terminals one by one, and maintains a stack $\alpha \in \Pi^+$, where $\Pi$ is the stack alphabet. Its action is determined by the stack $\alpha$ and the current input terminal $a \in \Gamma$. (1) If the stack top $\alpha_{[0]}$ belongs to the *final states* $F_{\mathcal{P}}$, the parser terminates and *accepts* the input. (2) If the top 2 symbols $\alpha_{[:2]}$ can perform an $\varepsilon$-transition, i.e., $\alpha_{[:2]} \xrightarrow{\varepsilon}_{\mathcal{P}} \beta \in \Pi^+$, the parser pops $\alpha_{[:2]}$ to push $\beta$ onto the stack. (3) If the top 2 symbols $\alpha_{[:2]}$ can perform a transition for the input terminal $a$, i.e., $\alpha_{[:2]} \xrightarrow{a}_{\mathcal{P}} \beta$, the parser pops $\alpha_{[:2]}$ to push $\beta$ onto the stack, and then reads in the terminal $a$. (4) Otherwise, the parser terminates and *rejects* the input.

A parser stack $\alpha$ is *stable* if it is ready to read the next terminal or accepts the input, i.e., actions (1) and (3) above. We write $\alpha \xRightarrow{w}_{\mathcal{P}}^* \beta$ if $\mathcal{P}$ processes the terminal sequence $w \in \Gamma^*$ and transits from stack $\alpha$ to a *stable* stack $\beta$. The parser is *deterministic* if at any time the parser has only one possible action. It is *terminating* if for any stack, it does not make an endless sequence of $\varepsilon$-transitions.

### 2.4 RELATED WORK

There are several types of existing techniques to speed up grammar-constrained decoding: vocabulary preprocessing, lexer preprocessing, and parser preprocessing.

---

[1]To ease the presentation, the step of lexical analysis is omitted in previous sections. The term "parser" in previous sections should be realized as the combination of the lexer and the parser defined here, and the term "parser stack" should be realized as the concatenation of the lexer state and the parser state, which is a simple state without internal structure, and a stack, respectively.

**Vocabulary preprocessing** (Poesia et al.; Beurer-Kellner et al.; Moskal et al.) exploits the fact that the vocabulary is built by BPE (Gage; Sennrich et al.), and for each token, its prefix is also a token in the vocabulary. If the prefix token is rejected, then the longer token must also be rejected. So we can check the vocabulary hierarchically, and only check the tokens whose prefixes are not rejected.

**Lexer preprocessing** (Beurer-Kellner et al.; Park et al.; Moskal et al.; Ugare et al.) maps each token to a terminal sequence during preprocessing, and then the parser is only called on the terminal sequences. This reduces the number of parser calls, as different tokens may share the same terminal sequence. The mask can be precomputed for each terminal sequence, combined at runtime to get the valid token mask. Syncode (Ugare et al.) further approximates the terminal sequences by only considering the first 2 terminals of each token, removing the need for dynamic parsing using the lookaheads of the LR(1) parser at the cost of allowing certain invalid tokens to be accepted.

**Parser preprocessing** (Dong et al.; Park et al.) classifies the vocabulary into three sets for each parser state: context-independent accepted, context-independent rejected, and context-dependent. This allows us to reduce the number of parser calls by only checking the context-dependent tokens.

These techniques can be combined to achieve better speedup (Beurer-Kellner et al.; Park et al.; Moskal et al.). However, as mentioned in Section 1, these techniques are limited in their speedup while maintaining correctness. There is no theoretical guarantee on how many parser calls will be made per decoding step, which can be linear to the vocabulary size in the worst case.

## 3 PSC: PARSER STACK CLASSIFICATION

The preprocessing of the lexer $\mathcal{T}$ is described in Section 3.1, and the other parts of this section are dedicated to the preprocessing of the parser $\mathcal{P}$. Detailed algorithms and proofs are deferred to Appendix A.3. **A quick lookup table for the symbols, notations, and the exact definitions can be found in Appendix A.2.**

### 3.1 LEXICAL PREPROCESSING

Lexical preprocessing is not our focus in this paper, so we reuse the lexical preprocessing in Great-Gramma (Park et al.), and conclude it here as a prelude to PSC.

For token $v \in \mathcal{V}$, lexer state $q \in Q$, if lexing the token $v$ from state $q$ using the lexer $\mathcal{T}$ generates the terminal sequence $x \in \Gamma^*$, and the lexer transits to state $p$, i.e. $q \xrightarrow[\mathcal{T}]{v:x}^* p$, we define the set of *realizable terminal sequences* $R_q(v)$ as $\{x\}T_p$, representing all possible terminal prefixes that can be generated from a string starting with $v$, where $T_p$ is the finite set of all possible terminal prefixes from state $p$: $\mathcal{T}_p(\Sigma^*) = T_p\mathcal{T}(\Sigma^*)$. The set $R_q(v)$ can be precomputed for every $q \in Q, v \in \mathcal{V}$ during preprocessing.

### 3.2 OVERVIEW OF SYNTACTIC PREPROCESSING

Given a valid prefix $x \in \Sigma^*$, we run the lexer $\mathcal{T}$ from its initial state $q_0$ to produce a terminal sequence $z \in \Gamma^*$ and a new lexer state $q$: $q_0 \xrightarrow[\mathcal{T}]{x:z}^* q$. We then run the parser $\mathcal{P}$ from its initial stack $\gamma_0$ to receive the terminal sequence $z$, and generates a new stack $\alpha$: $\gamma_0 \xrightarrow[\mathcal{P}]{z}^* \alpha$.

We can now introduce the simplification of the condition in the GCD definition in Equations 1 and 2 **from previous work (Park et al.).** For any token $v \in \mathcal{V}$, to determine whether $v$ is valid, we can rewrite the condition in terms of realizable terminal sequences,

$$\exists y \in \Sigma^*, \mathcal{T}(xvy) \in \mathcal{P} \iff \exists w \in R_q(v), \exists \beta \in \Pi^+, \alpha \xrightarrow[\mathcal{P}]{w}^* \beta. \tag{3}$$

The simplification is based on the common assumption that, if the parser reads in a certain terminal sequence and enters a stable stack, then we do not need to worry about the rest of the input, and there always exists a terminal sequence produced by the lexer that can ensure the whole text is accepted.

For $w \in \Gamma^*$, we define $P_w(\alpha)$ for the calculation in the last step of Equation 3,

$$P_{w \in \Gamma^*}(\alpha \in \Pi^+) := \left\{ \beta \in \Pi^+ \middle| \alpha \xrightarrow[\mathcal{P}]{w}^* \beta \right\}. \tag{4}$$

**How to efficiently calculate $P_w(\alpha)$ is the key difference between PSC and previous methods.** In existing work, the calculation of $P_w$ is almost always dynamic: one has to calculate $P_w(\alpha)$ for the current $\alpha$ and every possible $w \in R_q(\mathcal{V})$. While existing methods in Section 2.4 employ precomputation to optimize certain cases, they still fundamentally require worst-case $\mathcal{O}(|R_q(\mathcal{V})|)$ time for dynamic parsing if correctness is not sacrificed.

In this work, PSC proposes a totally different approach, modeling $P_w$ as a deterministic finite-state transducer (FST), which reads the stack sequence $\alpha \in \Pi^+$, and then outputs the sequence $\beta \in \Pi^+$ if there is one, or rejects the input stack otherwise.

This gives us several benefits. Because each $P_w$ reads and outputs a sequence of stack symbols, they can be composed to create larger FSTs: $P_t \circ P_s = P_{st}$. Because the realizable terminal sequences are known during precomputation, the exact validity condition of each vocabulary is therefore known, their combinations can be precomputed, and we only need to go through the current stack once during runtime. All possible masks can also be precomputed, eliminating the mask generation overhead during decoding.

The challenge here is whether and how each $P_w$ can be constructed as a deterministic FST. This is not straightforward because of the presence of $\varepsilon$-transitions in the PDA $\mathcal{P}$. To address this, we first construct $P_\varepsilon$ to handle all $\varepsilon$-transitions, and then construct $P_w$ for any $w \in \Gamma^*$ based on $P_\varepsilon$.

### 3.3 FST OF $\varepsilon$ TRANSITIONS

In this section, we construct the FST $P_\varepsilon$. Its input should be a stack, and **the output is its stabilized version, by repeatedly executing all needed $\varepsilon$ transitions on the stack**. The start state is $\varepsilon$, and the final state is a special state FINAL. The set of all states is the minimum closure of the transitions defined below, where each state represents the current known stack top. In Appendix A.3.1, we give a proof that this is a finite set, thus forming a finite-state transducer (FST).

$$\forall X \in \Pi, \qquad \alpha \quad \xrightarrow[P_\varepsilon]{X:\varepsilon} \quad \alpha X, \qquad \text{if } |\alpha| < 2 \text{ and } \alpha_{[0]} \notin F_\mathcal{P};$$
$$\alpha \quad \xrightarrow[P_\varepsilon]{\varepsilon:\alpha} \quad \text{FINAL}, \quad \text{if } \alpha_{[:2]} \xrightarrow[\mathcal{P}]{a} \beta, \exists a \in \Gamma \text{ or } \alpha_{[0]} \in F_\mathcal{P};$$
$$\alpha \quad \xrightarrow[P_\varepsilon]{\varepsilon:\varepsilon} \quad \beta\alpha_{[2:]}, \quad \text{if } \alpha_{[:2]} \xrightarrow[\mathcal{P}]{\varepsilon} \beta;$$
$$\forall X \in \Pi, \quad \text{FINAL} \quad \xrightarrow[P_\varepsilon]{X:X} \quad \text{FINAL}.$$

There are four types of transitions in $P_\varepsilon$. (1) If one cannot determine whether the stack is stable from the stack top $\alpha$, it transits to a new state by reading the next stack symbol. (2) If the stack top $\alpha$ is stable, it transits to the FINAL state to output the final stable stack. (3) Otherwise, it simulates the transition of $\mathcal{P}$ on the current stack top $\alpha$, and transits to a new state representing the new stack top after executing the $\varepsilon$ transition. (4) In the FINAL state, it always outputs the input stack unchanged.

$P_\varepsilon$ is an important building block in the construction of other $P_w$, $w \in \Gamma^+$. For any stack $\alpha$, $P_\varepsilon(\alpha)$ gives the stabilized version of $\alpha$, so the FST composed after $P_\varepsilon$ does not need to handle $\varepsilon$ transitions, and we can compose $P_\varepsilon$ after other FSTs to meet the stability requirement in the definition of $P_w$.

### 3.4 FST FOR ANY TERMINAL SEQUENCE

After constructing $P_\varepsilon$, the construction of $P_w$ for any terminal sequence $w \in \Gamma^+$ is fairly simple.

We first construct an FST $\tilde{P}_a$ for every $a \in \Gamma$ that **simulates a single transition labeled** $a$, i.e. outputting $\beta$ for the input stack $\alpha$ if $\alpha \xrightarrow[\mathcal{P}]{a} \beta$. Note that the output stack is not required to be stable. The start state is $\varepsilon$, the final state is FINAL, and the transitions are defined as follows.

$$\varepsilon \xrightarrow[\tilde{P}_a]{X:\varepsilon} X \xrightarrow[\tilde{P}_a]{Y:\varepsilon} XY \xrightarrow[\tilde{P}_a]{\varepsilon:\beta} \text{FINAL}, \forall XY \xrightarrow[\mathcal{P}]{a} \beta; \qquad \text{FINAL} \xrightarrow[\tilde{P}_a]{X:X} \text{FINAL}, \forall X \in \Pi.$$

For any terminal sequence $w = w_1 \ldots w_n \in \Gamma^+$, the FST $P_w$ can be constructed as follows.

$$P_w = P_\varepsilon \circ \tilde{P}_{w_1} \circ P_\varepsilon \circ \cdots \circ P_\varepsilon \circ \tilde{P}_{w_n} \circ P_\varepsilon. \tag{5}$$

Intuitively, the input stack $\alpha$, is first passed to $P_\varepsilon$ to get a stable stack, and then passed to $\tilde{P}_{w_1}$ to get the stack after reading $w_1$, and then passed to $P_\varepsilon$ to get the stabilized version, etc, until it is passed to $\tilde{P}_{w_n}$ and stabilized with $P_\varepsilon$. Relevant algorithms and proofs are given in Appendix A.3.2.

When calculating the mask, we only care about whether $P_w(\alpha) \neq \varnothing$. Removing all the output labels from $P_w$ gives us a finite-state automaton, hereafter named $A_w$.

## 3.5 ONE-PASS FSA FOR MASK SELECTION

After constructing $P_w$ for all realizable terminal sequences $w \in R(\mathcal{V})$, we can now consider simplifying the constraint calculation over the whole vocabulary $\mathcal{V}$. Recall Equation 1, combined with Equation 3 and $A_w$, we have the following equation,

$$c(x \in \Sigma^*) = \{v \in \mathcal{V} | \exists w \in R_q(v), P_w(\alpha) \neq \varnothing\} = \{v \in \mathcal{V} | \exists w \in R_q(v), \alpha \in A_w\},$$

where $q$ and $\alpha$ as defined in Section 3.2 are only dependent on $x$.

In $c(x)$, we want to know whether any of the $A_w$ accepts $\alpha$, where $w \in R_q(v)$. This can be achieved by constructing the union of different $A_w$, i.e., $\bigcup_{w \in R_q(v)} A_w$.

For different tokens $v \in \mathcal{V}$, we need to check whether $\alpha$ is accepted by $\bigcup_{w \in R_q(v)} A_w$. But to get the whole mask $c(x)$, we need to check for every $v \in \mathcal{V}$, which is inefficient. To address this issue, we can integrate the checking of $v$ and $q$ into the FSA. Introduce the notation $B_a$ for an FSA that accepts only $a$ once. For every $q \in Q$ and $v \in \mathcal{V}$, we can concatenate $B_q$ before $\bigcup_{w \in R_q(v)} A_w$ to check whether the current state is $q$, and then concatenate $B_v$ after $\bigcup_{w \in R_q(v)} A_w$ to check whether the candidate token is $v$.

By unioning the results for all $v \in \mathcal{V}$ and $q \in Q$, we construct an FSA $\mathcal{A}$ that accepts the sequence $q\alpha v$ only if $v$ is a valid token for the lexer state $q$ and stack $\alpha$,

$$\mathcal{A} := \bigcup_{v \in \mathcal{V}} \bigcup_{q \in Q} \bigcup_{w \in R_q(v)} B_q A_w B_v, \quad c(x) = \{v \in \mathcal{V} | q\alpha v \in \mathcal{A}\}, \tag{6}$$

where $\mathcal{A}$ should be determinized and minimized. This gives us the following theorem.

**Theorem 1.** *All (lexer state, parser stack) pairs that accept a given token form a regular language.*

In Equation 6, we can precompute all possible result of $c$, i.e. all possible vocabulary masks, by considering acceptable vocabulary set $\mathcal{A}_s := \left\{ v \in \mathcal{V} \middle| s \xrightarrow{v}_{\mathcal{A}} f^{\mathcal{A}} \right\}$ for every state $s$ in $\mathcal{A}$ where $f^{\mathcal{A}}$ is the final state of $\mathcal{A}$.

We summarize the offline construction process of PSC in Algorithm 1, and the online execution process in Algorithm 2. In Algorithm 2, both the lexing step 2 and the parsing step 3 are standard in grammar-constrained decoding, and can be incrementally maintained. In Step 4, the FSA $\mathcal{A}$ is run on the stack $\alpha$ and the lexer state $q$ to get the state $s$, only requiring $\mathcal{O}(|\alpha|)$ time. Step 5 can be precomputed to be $\mathcal{O}(1)$ at runtime.

---

**Algorithm 1** Offline construction in PSC

1: **function** OFFLINECONSTRUCTION($\mathcal{T}, \mathcal{P}, \mathcal{V}$)
2:      $P_\varepsilon \leftarrow$ EPSILONFST($\mathcal{P}$)
3:      **for all** $a \in \Gamma$ **do**
4:         $\tilde{P}_a \leftarrow$ TERMINALFST($\mathcal{P}, a$)
5:      **for all** $w = w_1 \ldots w_n \in R(\mathcal{V})$ **do**
6:         $P_w \leftarrow P_\varepsilon \circ \tilde{P}_{w_n} \circ P_\varepsilon \circ \cdots \circ P_\varepsilon \circ \tilde{P}_{w_1} \circ P_\varepsilon$
7:         $A_w \leftarrow$ REMOVEOUTPUT($P_w$)
8:      $\mathcal{A} \leftarrow \bigcup_{v \in \mathcal{V}} \bigcup_{q \in Q} \bigcup_{w \in R_q(v)} B_q A_w B_v$
9:      $\mathcal{A} \leftarrow$ MINIMIZE($\mathcal{A}$)
10:     **return** $\mathcal{A}$

**Algorithm 2** Online execution of PSC

1: **function** ONLINEEXECUTION($\mathcal{T}, \mathcal{P}, \mathcal{A}, x$)
2:      $q_0^{\mathcal{T}} \xRightarrow[\mathcal{T}]{x:z} {}^* q$
3:      $\gamma_0 \xRightarrow[\mathcal{P}]{z} {}^* \alpha$
4:      $q_0^{\mathcal{A}} \xRightarrow[\mathcal{A}]{q\alpha} {}^* s$
5:      **return** $\mathcal{A}_s$

---

## 4 EXPERIMENTS

### 4.1 EXPERIMENTAL SETUP

All grammar-constrained decoding methods essentially perform the same task: compute the valid token mask at each decoding step. **The valid token mask is theoretically the same for all methods, so we focus on comparing the efficiency of mask computation in our experiments.** The usefulness of grammar-constrained decoding is shown in previous work (Geng et al., b; Scholak et al.; Poesia et al.; Ugare et al.); nevertheless, we replicate the downstream task performance of PSC (which is the same as other GCD methods) in Appendix A.6.

We conduct two sets of experiments to evaluate each method:

- **Overhead of mask computation (without model inference).**
- **End-to-end throughput (with model inference).**

In all experiments, we use **teacher-forcing** during evaluation, i.e., we always use the oracle next token at each decoding step, **to ensure that all methods are evaluated under the same conditions and can be fairly compared.**

**Datasets** There is no standard benchmark for evaluating grammar-constrained decoding methods. We choose two representative tasks that require grammar-constrained decoding: code generation in Java, Go, and SQL, and JSON generation with specified JSON schemas. For each task, we construct the evaluation dataset as described in Appendix A.4, with 1000 samples for each programming language and 1000 schemas for schema-conformant JSON generation. There are a total of 1337 positive samples and 2072 negative samples in the JSON dataset, and each schema has at least one positive sample and one negative sample.

**Baselines** We consider several recent state-of-the-art grammar-constrained decoding methods with open-source implementations as baselines, including XGrammar (Dong et al.), GreatGramma (Park et al.), Formatron (Sun et al.), and LLGuidance (Moskal et al.). Detailed descriptions of these baselines are included in Appendix A.5.

**Implementation** We implement PSC in roughly 1100 lines of Python. Similar to previous work (Ugare et al.; Park et al.), we use the Lark parser(lar) to construct the lexer and the LALR(1) parser from the grammar. As described in Section 3.1, we reuse the lexer construction in Great-Gramma (Park et al.), since this is not our focus in this paper.

**Execution environment** We conduct our experiments on a machine with 8 NVIDIA A100 GPUs (40 GB Memory), 2 Intel Xeon Gold 6348 CPUs (2.6GHz, 56 cores), and 512 GB RAM. To ensure fairness, we run all the experiments with a single GPU and a single CPU thread.

### 4.2 OVERHEAD OF MASK COMPUTATION

**Metrics** In this experiment, we measure the overhead of mask computation for each GCD method. We ignore the time taken to transfer the mask to the GPU and apply it to the logits, because this is the same for all methods[2]. For each method, we measure:

- **Average overhead:** The time of computing the CPU mask tensor at each decoding step.
- **Sample pass rate:** The proportion of samples that are correctly processed by each method[3].

**Models** We evaluate the overhead on three open-source LLM series with different vocabulary sizes: Llama 3 (Grattafiori et al.) (128k vocabulary size), Qwen 2.5 (Bai et al.) (151k vocabulary size), and Gemma 3 (Gemma Team et al.) (262k vocabulary size). We only use the tokenizers, because the overhead of mask computation is independent of other model components.

---

[2]In PSC, one can preload all the mask tensors on the GPU memory before decoding begins, eliminating the transfer overhead. However, the transfer overhead is usually too small to justify the extra GPU memory usage.

[3]If the oracle token is masked, the sample is counted as rejected by the method. A positive sample is counted as passed if it is not rejected, and a negative sample is counted as passed if it is rejected.

Table 1: Average overhead (microseconds) of computing the mask per token in GCD tasks using different methods. The symbol X indicates the parser reports an error during mask calculation. Text in **bold** indicates the best performance, and text in underline indicates the second best performance.

| Model | Method | Grammar | | | |
|---|---|---|---|---|---|
| | | Java | Go | SQL | JSON Schemas |
| Llama 3 $\|\mathcal{V}\| = 128256$ | XGrammar | 309514.2 | 281500.9 | 324663.6 | 26257.6 |
| | Formatron | 393540.4 | X[a] | 303974.1 | X[a] |
| | GreatGramma | 21402.8 | 27220.9 | 20954.5 | 8556.2 |
| | LLGuidance | 1352.4 | X[b] | 826.1 | 72.7 |
| | PSC (Ours) | **2.4** | **2.5** | **2.6** | **2.3** |
| Qwen2.5 $\|\mathcal{V}\| = 151665$ | XGrammar | 302421.9 | 278333.4 | 299968.6 | 29470.0 |
| | Formatron | 378921.0 | X[a] | 253311.9 | X[a] |
| | GreatGramma | 24570.9 | 27649.8 | 24053.9 | 11038.4 |
| | LLGuidance | 1408.2 | X[b] | 865.2 | 72.1 |
| | PSC (Ours) | **2.4** | **2.5** | **2.5** | **2.2** |
| Gemma 3 $\|\mathcal{V}\| = 262145$ | XGrammar | 649321.1 | 458952.0 | 416625.0 | 54164.4 |
| | Formatron | 696144.6 | X[a] | 444810.0 | X[a] |
| | GreatGramma | 43218.6 | 48354.0 | 40026.9 | 25717.3 |
| | LLGuidance | 1802.6 | X[b] | 1180.8 | 72.1 |
| | PSC (Ours) | **2.3** | **2.5** | **2.4** | **2.3** |

[a] Formatron stops responding on multiple samples, so we terminate the process.
[b] LLGuidance reports `ParserTooComplex` error.

**Overhead results**  The average overhead is shown in Table 1. PSC significantly outperforms all the baselines across all grammars and models, being 310 to 700 times faster on complex programming language grammars compared to the best baseline, LLGuidance, and generally 30 times faster on the relatively simple JSON schema grammar.

The overhead of baselines is significantly larger on Gemma 3 than that on the other models[4], because Gemma 3 has a much larger vocabulary size (262k) than the other two models (128k and 151k), and the time complexity of all methods except PSC is roughly linear to the vocabulary size. In contrast, the performance of PSC is stable across different grammars and models, because its time complexity is independent of the vocabulary size.

**Sample pass rate results**  The sample pass rates are shown in Table 2. PSC achieves a high pass rate across all grammars and models, very close to 100%. After analyzing the error cases, we find that they are all directly rejected by the GreatGramma lexer we adopt, but **no error is caused by PSC itself**. We notice the strangely low sample pass rate of Formatron on SQL, and find that Formatron refuses to accept the column alias in the query, resulting in frequent rejections.

## 4.3  End-to-end Throughput

**Settings**  In this experiment, we run the actual model inference using the vLLM library (Kwon et al.), and measure the throughput, i.e., the number of tokens processed per second, of the entire decoding process, considering both the time taken by mask computation and model inference. We only consider the accepted samples of each method. We compare PSC with the fastest baseline LLGuidance in the previous section, and also include the throughput when not using any constraint decoding method as a reference, under various batch sizes of 1, 2, 4, ..., 256.

**Models**  We use the smallest models in the three model series to highlight the overhead of constraint decoding. On these models, the model inference time is relatively small, making the over-

---

[4]The performance of LLGuidance on JSON Schemas is roughly the same across different models, probably because the grammar is simple enough that the vocabulary size does not significantly affect the performance.

Table 2: How many samples are correctly processed by each method. The symbol X indicates the parser reports an error during mask calculation. Text in **bold** indicates the best performance, and text in underline indicates the second best performance.

| Model | Method | Grammar | | | |
| | | Java | Go | SQL | JSON Schemas |
|---|---|---|---|---|---|
| Llama 3 | XGrammar | 99.7% | **100.0%** | 99.7% | **100.0%** |
| | Formatron | 99.4% | X | 67.0% | X |
| | GreatGramma | **100.0%** | 99.9% | 97.1% | 99.6% |
| | LLGuidance | **100.0%** | X | **99.9%** | 99.9% |
| | PSC (Ours) | **100.0%** | 99.9% | **99.9%** | 99.6% |
| Qwen2.5 | XGrammar | 99.7% | **100.0%** | 99.7% | **100.0%** |
| | Formatron | 99.4% | X | 67.0% | X |
| | GreatGramma | **100.0%** | 99.9% | 97.2% | 99.6% |
| | LLGuidance | **100.0%** | X | **99.9%** | 99.9% |
| | PSC (Ours) | **100.0%** | 99.9% | **99.9%** | 99.6% |
| Gemma 3 | XGrammar | 99.7% | **100.0%** | 99.7% | **100.0%** |
| | Formatron | **100.0%** | X | 67.3% | X |
| | GreatGramma | **100.0%** | 99.9% | 97.2% | 99.6% |
| | LLGuidance | 94.0% | X | **99.9%** | 99.9% |
| | PSC (Ours) | **100.0%** | 99.9% | **99.9%** | 99.6% |

head of constraint decoding more pronounced. Specifically, we use Llama 3 1B, Qwen 2.5 0.5B, and Gemma 3 270M. We also include Qwen 2.5 7B to see the effect of model size on throughput.

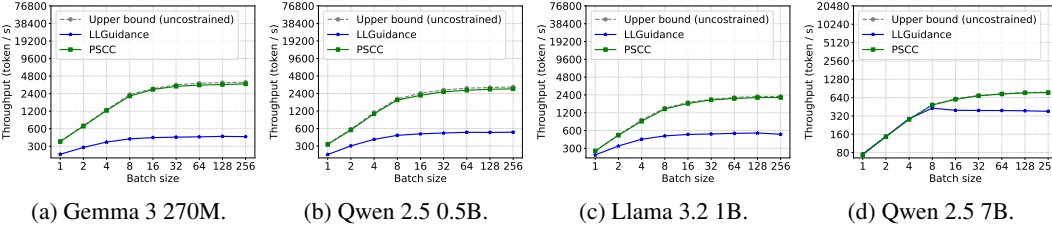

(a) Gemma 3 270M.  (b) Qwen 2.5 0.5B.  (c) Llama 3.2 1B.  (d) Qwen 2.5 7B.

Figure 2: End-to-end throughput (tokens per second) on the **Java** dataset using different methods on different models with various batch sizes.

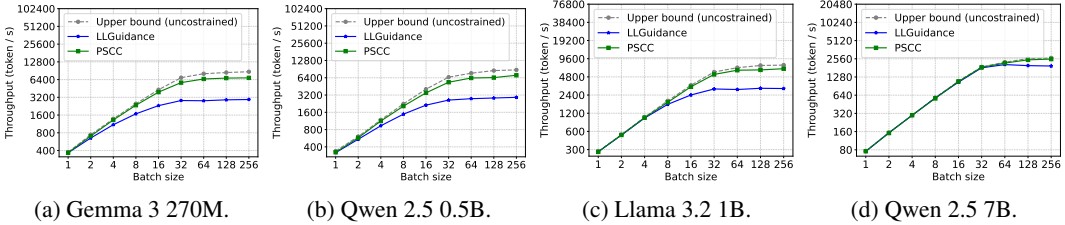

(a) Gemma 3 270M.  (b) Qwen 2.5 0.5B.  (c) Llama 3.2 1B.  (d) Qwen 2.5 7B.

Figure 3: End-to-end throughput (tokens per second) on the **schema-conformant JSON** dataset using different methods on different models with various batch sizes.

**Results** The end-to-end throughput results on Java and schema-conformant JSON datasets are present in Figures 2 and 3, respectively. The results on the Go and SQL datasets are similar to those on the Java dataset, included in Appendix A.7.

On all datasets, PSC consistently outperforms LLGuidance across all models and batch sizes, and is very close to the performance of unconstrained decoding. The difference is more pronounced on the Java dataset, where the grammar is more complex, leading to higher overhead for LLGuidance.

Table 3: Preprocessing overhead of PSC on different grammars.

| Metrics | Model | Grammar | | | |
|---|---|---|---|---|---|
| | | Java | Go | SQL | JSON Schemas |
| Time (seconds) | Llama 3 | 466.9 | 1171.7 | 4662.7 | 28.3 |
| | Qwen2.5 | 464.5 | 1166.8 | 4770.0 | 28.6 |
| | Gemma 3 | 472.1 | 1362.2 | 1367.4 | 53.2 |
| Memory (GiB) | Llama 3 | 40.8 | 87.7 | 255.3 | 3.04 |
| | Qwen2.5 | 40.4 | 86.6 | 254.2 | 3.13 |
| | Gemma 3 | 36.0 | 88.8 | 188.7 | 5.95 |
| Disk Space (MiB) | Llama 3 | 13.27 | 28.16 | 58.95 | 0.54 |
| | Qwen2.5 | 12.70 | 28.37 | 57.60 | 0.51 |
| | Gemma 3 | 13.13 | 33.52 | 24.04 | 0.76 |

As the model size increases, the difference in throughput becomes smaller, because the model inference time becomes more dominant. However, since smaller models are less capable, grammar-constrained decoding is probably more useful for smaller models to ensure the syntactic correctness.

As the batch size increases, the difference in throughput becomes larger, because the average model inference time per token decreases with larger batch sizes, indicating that the overhead introduced by constraint decoding becomes more pronounced at larger batch sizes.

## 5    DISCUSSION

In this section, we discuss the preprocessing overhead of PSC, including the time, memory footprint of preprocessing, and the disk usage of preprocessing results. We compress the preprocessing results to disk with zstandard (Collet & Kucherawy, 2021) to avoid redundant preprocessing.

The preprocessing overhead is presented in Table 3. For JSON schemas, the average preprocessing time is around half to one minute per schema, and the memory footprint is around 3 GiB for Llama 3 and Qwen2.5, and around 6 GiB for Gemma 3. The disk usage is around half to one megabyte after compression. This is quite practically reasonable, allowing for quick adaptation to new schemas.

For the programming language grammars, the preprocessing time ranges from around 8 minutes for Java to around 1.3 hours for SQL. The memory footprint ranges from around 40 GiB for Java to around 250 GiB for SQL. The disk usage is generally tens of megabytes after compression.

While the preprocessing overhead for programming language grammars is higher than that for JSON schemas, it is still acceptable as it only needs *once* per grammar and vocabulary pair, and the grammars of programming languages are typically stable. The preprocessing results can be redistributed, so users can be free from the preprocessing overhead. The memory footprint, although high, is still feasible on modern machines with large memory capacity, and it can be done on cloud instances if needed. Also, the memory usage can potentially be reduced by implementation optimization. See Appendix A.8 for more discussion on the balance between preprocessing and runtime overhead.

Overall, the preprocessing overhead is generally manageable for practical applications.

## 6    CONCLUSION

In this paper, we present PSC, a novel approach for grammar-constrained decoding. By constructing the exact requirements on the parser stack for each vocabulary token, PSC can determine the valid tokens at each decoding step by a single pass through the parser stack. Our experimental results demonstrate that PSC achieves significant speedup over existing methods, and the end-to-end throughput of PSC approaches that of unconstrained decoding. This makes PSC a practical choice for real-world applications that require grammar-constrained decoding. In future, we plan to explore other types of grammars and constraints that can be efficiently handled by PSC, as well as other optimizations to further improve its efficiency and scalability.

## 7 REPRODUCIBILITY STATEMENT

We open-source the code to facilitate reproducibility of our results at `https://anonymous.4open.science/r/PSC-E43E`. It includes the implementation of PSC, the datasets used in the experiments, the scripts to run the experiments and generate the results in the paper, and the preprocessing results built from the grammars used in our experiments. The proofs of statements in the main text are included as Appendix A.3. The details in dataset construction are included as Appendix A.4.

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

# A APPENDIX

## A.1 LLM USAGE

We used DeepSeek and GitHub Copilot for code assistance, paper writing, and proofreading. However, all the technical content, ideas, algorithms, and experimental results in this paper are our own work. We carefully reviewed and verified all the content generated by LLMs to ensure they are accurate and directly reflect our own ideas.

## A.2 FORMAL DEFINITIONS AND NOTATIONS

Table 4: Symbols and their meaning.

| Symbol | Meaning |
|--------|---------|
| $\varepsilon$ | empty string |
| $ab$ | concatenation of strings $a$ and $b$ |
| $AB$ | concatenation of languages $A$ and $B$ |
| $A_\varepsilon$ | $A \cup \{\varepsilon\}$ |
| $A^*$ | Kleene star of language $A$ |
| $A^+$ | $A^* \setminus \{\varepsilon\}$ |
| $\Sigma$ | the character set (usually the Unicode) used by the language model |
| $\Gamma$ | the terminal set of the grammar |
| $\mathcal{V}$ | the vocabulary of the language model, a finite subset of $\Sigma^+$ such that every string over $\Sigma$ can be tokenized as a string over $\mathcal{V}$ |
| $\mathcal{T}$ | the lexing FST, transduces string over $\Sigma$ to terminal sequence over $\Gamma$ |
| $Q$ | the finite set of states of the lexing FST |
| $\mathcal{P}$ | the parsing PDA, accepts terminal sequences in the language |
| $\Pi$ | the stack alphabet of the PDA |

We include a list of symbols and their meanings in Table 4 for reference.

### A.2.1 FINITE-STATE TRANSDUCER (FST)

A *finite-state transducer* (FST) (Aho & Ullman) $\mathcal{T}$ is defined by a finite set of states $Q$, the input alphabet $\Sigma$, the output alphabet $\Gamma$, the start state $q_0 \in Q$, the final states $F \subseteq Q$, and transitions $\delta : Q \times \Sigma_\varepsilon \to 2^{\Gamma^* \times Q}$. If $\delta(q, a) \ni (y, q')$, we write $q \xrightarrow[\mathcal{T}]{a:y} q'$. We write $\to^*$ for consecutive transitions. For $q \in Q$, we write $q \xrightarrow[\mathcal{T}]{\varepsilon:\varepsilon}{}^* q$. For $q \xrightarrow[\mathcal{T}]{s:x}{}^* q'$ and $q' \xrightarrow[\mathcal{T}]{t:y}{}^* q''$, we write $q \xrightarrow[\mathcal{T}]{st:xy}{}^* q''$.

Informally, an FST is *deterministic*, if from any state, for any given string, there is exactly one possible outcome. $\mathcal{T}$ is *deterministic* if, for all $q \in Q$, either $|\delta(q, a)| \leq 1, \forall a \in \Sigma$ and $\delta(q, \varepsilon) = \varnothing$, or $\delta(q, a) = \varnothing, \forall a \in \Sigma$ and $\delta(q, \varepsilon) = 1$.

For $w \in \Sigma^*, q \in Q$, we define $\mathcal{T}_q(w)$ as $\left\{ v \in \Gamma^* \middle| \exists q' \in F, q \xrightarrow[\mathcal{T}]{w:v}{}^* q' \right\}$, meaning the possible outcomes when we feed $w$ into $\mathcal{T}$ starting from the state $q$. When $\mathcal{T}$ is deterministic and $v \in \mathcal{T}(w)$, we also write $\mathcal{T}(w) = v$. For $W \subseteq \Sigma^*$, we define $\mathcal{T}_q(W)$ as $\bigcup_{w \in W} \mathcal{T}_q(w)$. $q$ defaults to $q_0$ when omitted.

We call a state $q \in Q$ *stable* if the FST does not need to take any immediate action on $q$, i.e. $\delta(q, \varepsilon) = \varnothing$. If $q \xrightarrow[\mathcal{T}]{s:t}{}^* q'$ and $q'$ is stable, we also write $q \xRightarrow[\mathcal{T}]{s:t}{}^* q'$.

Given two FSTs $\mathcal{S}$ and $\mathcal{T}$ where the output alphabet of $\mathcal{S}$ is the input alphabet $\mathcal{T}$, $\Gamma^\mathcal{S} = \Sigma^\mathcal{T}$, their *composition* is a new FST $\mathcal{S} \circ \mathcal{T}$ by feeding the output of $\mathcal{S}$ into the input of $\mathcal{T}$.

### A.2.2 FINITE-STATE AUTOMATON (FSA)

A *finite-state automaton* (FSA) $\mathcal{A}$ can be defined by removing all output labels from an FST. We say $\mathcal{A}$ *accept*s $w \in \Sigma^*$ from state $q$ if $\mathcal{A}_q(w) \neq \varnothing$, and write $w \in \mathcal{A}_q$, and $q$ defaults to $q_0$ when omitted. Two FSAs are *equivalent* if they accept the same language.

$\mathcal{A}$ is *deterministic* if there is no $\varepsilon$ transition in $\delta$. Every nondeterministic FSA can be *determinized* into an equivalent deterministic FSA (Hopcroft & Ullman), and every deterministic FSA can be *minimized* into an equivalent deterministic FSA with the smallest number of states (Hopcroft & Ullman).

The *union* of two FSAs $\mathcal{A}$ and $\mathcal{B}$ is a new FSA $\mathcal{A} \cup \mathcal{B}$ that accepts any sequence that is accepted by either $\mathcal{A}$ or $\mathcal{B}$.

The *concatenation* of two FSAs $\mathcal{A}$ and $\mathcal{B}$ is a new FSA $\mathcal{A}\mathcal{B}$ that accepts any sequence that can be split into two parts $x = yz$, where $\mathcal{A}$ accepts the first part $y$ and $\mathcal{B}$ accepts the second part $z$.

### A.2.3 PUSH-DOWN AUTOMATON (PDA)

A *push-down automaton* (PDA) (Aho & Ullman; Hopcroft & Ullman; Caucal & Monfort) $\mathcal{P}$ is defined by the input alphabet $\Gamma$, the stack alphabet $\Pi$, the initial stack $\gamma_0 \in \Pi^2$, the final states $F \subseteq \Pi$, and a finite set of transitions $\delta : \Pi^2 \times \Gamma_\varepsilon \to 2^{\Pi^+}$. Note that the definition here merges the states and the stack symbols in the traditional definition of PDA, but they are equivalent if we treat the stack top symbol as the state. If $\delta(\alpha, a) \ni \beta$, we write $\alpha \xrightarrow[\mathcal{P}]{a} \beta$. If $\alpha \xrightarrow[\mathcal{P}]{a} \beta$, for any $\gamma \in \Pi^*$, we also write $\alpha\gamma \xrightarrow[\mathcal{P}]{a} \beta\gamma$. We write $\to^*$ for consecutive transitions. For $\gamma \in \Pi^+$, we write $\gamma \xrightarrow[\mathcal{P}]{\varepsilon}^* \gamma$. If $\alpha \xrightarrow[\mathcal{P}]{a} \beta$, $\beta \xrightarrow[\mathcal{P}]{w}^* \gamma$, we write $\alpha \xrightarrow[\mathcal{P}]{aw}^* \gamma$.

Informally, a PDA is *deterministic*, if from any stack, for any given string, there is exactly one possible outcome. $\mathcal{P}$ is *deterministic* if, for any $\alpha \in \Pi^2$, either $|\delta(\alpha, a)| \leq 1, \forall a \in \Gamma$ and $\delta(\alpha, \varepsilon) = \varnothing$, or $\delta(\alpha, a) = \varnothing, \forall a \in \Gamma$ and $|\delta(\alpha, \varepsilon)| = 1$.

A deterministic PDA is *terminating*, if for any stack, it does not make an endless sequence of $\varepsilon$-input transitions.[5] Every deterministic PDA can be transformed into another equivalent deterministic terminating PDA (Sipser; Hopcroft & Ullman).

We call a state $\beta = \beta_1 \ldots \beta_n \in \Pi^+$ *stable* if $\beta_1$ is a final state, i.e. $\beta_1 \in F$, or the PDA is waiting to read one more symbol, i.e. $\exists a \in \Gamma, \delta(\beta_1\beta_2, a) \neq \varnothing$. If $\alpha \xrightarrow[\mathcal{P}]{w}^* \beta$ and $\beta$ is stable, we also write $\alpha \xRightarrow[\mathcal{P}]{w}^* \beta$. In practice, parser in a stable state is ready to consume the next input symbol, or has reached an accepting stack.

For $\alpha \in \Pi^+$, we define $\mathcal{P}_\alpha$ as $\left\{ w \in \Gamma^* \middle| \exists X \in F, \exists \gamma \in \Pi^*, \alpha \xRightarrow[\mathcal{P}]{w}^* X\gamma \right\}$. $\alpha$ defaults to $\gamma_0$ when omitted.

### A.3 ALGORITHMS AND PROOFS

In this section, we provide the exact algorithms of FST construction and proofs of their correctness.

---

[5]The definition is slightly different in the cited references; nevertheless, their proof works on this definition.

### A.3.1 FST OF $\varepsilon$ TRANSITIONS

---
**Algorithm 3** Construct FST $P_\varepsilon$

---
1: **function** EPSILONFST($\mathcal{P}$)
2:     **for all** $X \in \Pi$ **do**
3:         FINAL $\xrightarrow[P_\varepsilon]{X:X}$ FINAL
4:     $Q \leftarrow \{\varepsilon\}$
5:     **while let** $\alpha \in \Pi^*, Q \leftarrow \text{POP}(Q)$ **do**
6:         **if** $\alpha \in F_\mathcal{P}\Pi^*$ **then**
7:             $\alpha \xrightarrow[P_\varepsilon]{\varepsilon:\alpha}$ FINAL
8:         **else if** $|\alpha| < 2$ **then**
9:             **for all** $X \in \Pi$ **do**
10:                 $\alpha \xrightarrow[P_\varepsilon]{X:\varepsilon} \alpha X$
11:                 $Q \leftarrow Q \cup \{\alpha X\}$
12:         **else**
13:             **let** $\alpha_0\gamma = \alpha$,
                where $\alpha_0 \in \Pi^2, \gamma \in \Pi^*$
14:             **if** $\alpha_0 \xrightarrow{\varepsilon}{}_\mathcal{P} \beta$ **then**
15:                 $\alpha_0\gamma \xrightarrow[P_\varepsilon]{\varepsilon:\varepsilon} \beta\gamma$
16:                 $Q \leftarrow Q \cup \{\beta\gamma\}$
17:             **else if** $\alpha_0 \xrightarrow{a}{}_\mathcal{P} \beta$ **then**
18:                 $\alpha \xrightarrow[P_\varepsilon]{\varepsilon:\alpha}$ FINAL
19:     **return** $P_\varepsilon$

---

The algorithm for constructing the transitions of $P_\varepsilon$ is presented in Algorithm 3. We have the following theorem regarding the correctness of Algorithm 3.

**Theorem 2.** *Algorithm 3 constructs a finite-state transducer $P_\varepsilon$ as defined in Equation 4.*

*Proof.* First we show that the states of $P_\varepsilon$ are finite, i.e. Algorithm 3 terminates. Consider the two steps in Algorithm 3 that add new states. Step 11 can only adds states in $\Pi \cup \Pi^2$, so step 11 is only executed a finite number of times. As for Step 16, because the parser $\mathcal{P}$ is deterministic and terminating, by definition in Appendix A.2.3, for any stack configuration, there will not be an endless sequence of $\varepsilon$ transitions, so Step 16 is also only executed a finite number of times.

The correctness of Algorithm 3 can be naturally deduced by its construction, because it simply simulates the behavior of the parser $\mathcal{P}$ with the current stack top, and only outputs the stack when it is stable. $\qquad\square$

### A.3.2 FST OF ANY TERMINAL SEQUENCE

---
**Algorithm 4** Construct FST $\tilde{P}_a$ for terminal $a \in \Gamma$

---
1: **function** TERMINALFST($\mathcal{P}, a$)
2:     **for all** $X \in \Pi$ **do**
3:         FINAL $\xrightarrow[\tilde{P}_a]{X:X}$ FINAL
4:     **for all** $XY \xrightarrow{a}{}_\mathcal{P} \beta$ **do**
5:         $\varepsilon \xrightarrow[\tilde{P}_a]{X:\varepsilon} X \xrightarrow[\tilde{P}_a]{Y:\varepsilon} XY \xrightarrow[\tilde{P}_a]{\varepsilon:\beta}$ FINAL
6:     **return** $\tilde{P}_a$

---

The algorithm for constructing the transitions of $\tilde{P}_a$ is presented in Algorithm 4. We have the following theorem regarding the correctness of Algorithm 4 and Equation 5.

**Theorem 3.** *The above construction of $P_w$ meets the definition in Equation 4.*

*Proof.* Formally, $\tilde{P}_a$ can be defined as follows:

$$\tilde{P}_a(\alpha \in \Pi^+) := \left\{ \beta \in \Pi^+ \middle| \alpha \xrightarrow[\mathcal{P}]{a} \beta \right\}. \tag{7}$$

The construction of $\tilde{P}_a$ in Algorithm 4 trivially simulates one $a$-labelled transition of the parser $\mathcal{P}$ on the current stack top, so it meets Equation 7.

The process of the parser processing $w = w_1 \ldots w_n$ can be decomposed into a sequence of unconditional $\varepsilon$-transitions, followed by $w_i$-labelled transitions, followed by another sequence of unconditional $\varepsilon$-transitions. Each $w_i$-labelled transition is simulated by the corresponding $\tilde{P}_{w_i}$, and the unconditional $\varepsilon$-transitions are handled by $P_\varepsilon$. Therefore, the composition $P_w = P_\varepsilon \circ \tilde{P}_{w_1} \circ P_\varepsilon \circ \cdots \circ P_\varepsilon \circ \tilde{P}_{w_n} \circ P_\varepsilon$ correctly simulates the parser $\mathcal{P}$ processing the terminal sequence $w$ on the input stack $\alpha$, and produces the stabilized output stack $\beta$ if it exists. $\square$

### A.3.3 PROOF OF THEOREM 1

*Proof.* Because $\mathcal{A}$ is constructed as a FSA, the language recognized by $\mathcal{A}$ is regular. The language of all valid (lexer state, parser stack) pairs for a given token $v \in \mathcal{V}$ can be obtained by reversing the language recognized by $\mathcal{A}$, taking the Brzozowski derivative (Brzozowski, 1964) with respect to $v$, and then reversing it back. Since the class of regular languages is closed under these operations (Hopcroft & Ullman), the resulting language is also regular. $\square$

### A.4 DATASET CONSTRUCTION DETAILS

**Java, Go, and SQL** We obtain their Lark grammars from the previous work Syncode (Ugare et al.). Because the grammar format for XGrammar and Formatron is different from Lark, we manually convert the Lark grammars to respective formats for each baseline. For each programming language, we take the first 1000 samples from the Stack dataset (Kocetkov et al.) that can be successfully parsed by the Lark parser to construct the evaluation dataset.

**JSON Schemas** We use the benchmark dataset MaskBench (mas), an extension of JSON Schema Bench (Geng et al., a) by adding schema conformant and non-conformant JSON instances to each schema. We generate the Lark grammar from the JSON schemas using the script provided in MaskBench, and only use the schemas in MaskBench where the Lark parser can successfully parse all the conformant JSON instances and reject all the non-conformant JSON instances. We then randomly sample 1000 schemas for evaluation.

### A.5 DESCRIPTION OF BASELINES IN EXPERIMENTS

We describe the baselines used in our experiments in detail.

- XGrammar (Dong et al.) uses a character-level non-deterministic PDA[6]. For each state, it precomputes the context-independent accepted and rejected tokens, and only calls the parser for the context-dependent tokens.
- GreatGramma (Park et al.) uses a lexer and a parser. It converts each token into all possible terminals sequences and reduces the number of parser calls by sharing the parser calls among tokens with the same terminal sequence. After computing the accepted terminal sequences, it maps them back to the original tokens. It also precomputes the context-dependent and context-independent terminal sequences for each parser state.

---

[6]In the latest implementation that we use in the experiments, this has been changed to an Earley (Earley) parser.

- Formatron (Sun et al.) uses an Earley parser. It dynamically identifies and eliminates invalid or redundant parser states during parsing, and uses a state cache to speed up the repetitive parsing process.

- LLGuidance (Moskal et al.) uses a lexer and an Earley parser. It organizes the vocabulary into a trie, and skips the whole subtree if the prefix token is rejected. It also leverages the lexer on the vocabulary to pre-identify the terminal sequences.

## A.6 DOWNSTREAM PERFORMANCE OF GRAMMAR-CONSTRAINED DECODING

Our method PSC significantly speeds up the mask computation in grammar-constrained decoding. It calculates the same valid token masks as existing GCD methods, so its performance on downstream tasks should be similar to theirs, and we should observe similar downstream task performance improvements over unconstrained decoding. To verify this, we replicate the downstream task experiments from Syncode (Ugare et al.), comparing PSC with unconstrained decoding.

### A.6.1 JSON GENERATION

We replicate the schema-conformant JSON generation task from Syncode (Ugare et al.).

**Dataset** We use the JSON-Mode-Eval (NousResearch) dataset, which contains 100 samples of natural language instructions, each paired with a JSON schema and the corresponding correct JSON output. During checking, we found the oracle answer of sample 39 simply copies the schema (which is valid JSON but clearly not the intended output), so we exclude this sample from evaluation. When we generate the grammar from the JSON schema, we found that our script (obtained from MaskBench (mas), as described in Appendix A.4) fails to generate a valid grammar for certain schemas. To address this, we replace the schemas of sample 19, 24, 27, 33, 45 and 72 with the equivalent schemas supported by our grammar generation script; the schemas of sample 1, 15, 22, 90 and 97 cannot be converted to equivalent schemas supported by our grammar generation script, so we only enforce the JSON grammar on these samples. The exact schemas used in our experiments are provided in the open-sourced code.

**Settings** We compare three methods: unconstrained decoding (Standard), grammar-constrained decoding (with PSC) using only the JSON grammar (not specific to the schema) (GCD + JSON Grammar), and grammar-constrained decoding (with PSC) using the grammar generated from the JSON schema (GCD + JSON Schema). For the unconstrained decoding, we found that the generated JSON are often wrapped in code blocks (e.g. ```json ... ```), so we strip such code block markers before checking the validity of the generated JSON.

**Models** We use the instruct-tuned versions of Llama 3.2 1B, Qwen2.5 0.5B and Gemma 3 270M models. The same base models are used in our main experiments, and we use their instruct-tuned versions because the prompts contain chat-style instructions.

**Metrics** The generated JSON is considered correct if and only if it can be converted to a JSON object that exactly matches the oracle answer. For each sample, we generate 1 JSON output with greedy decoding, and generate 50 outputs with sampling (temperature 1.0, no top-$p$ or top-$k$ filtering). We report the pass@$k$ metric (Chen et al.) for $k = 1, 3, 5, 10, 20, 50$. For pass@1, we use the greedy decoding result; for pass@k where $k > 1$, we use the sampling results.

**Results** The results are presented in Table 5. For all the models tested, grammar-constrained decoding with PSC using the grammar generated from the JSON schema (GCD + JSON Schema) significantly outperforms unconstrained decoding (Standard) and grammar-constrained decoding using only the JSON grammar (GCD + JSON Grammar). This confirms that grammar-constrained decoding can effectively improve the performance of LLMs on schema-conformant JSON generation tasks, and PSC can achieve this improvement while significantly speeding up the existing GCD methods.

Table 5: The pass@$k$ scores (%) of different methods on generation of schema-conformant JSON using grammar-constrained decoding and unconstrained decoding (Standard). For the grammar-constrained decoding methods, we use PSC to compute the valid token masks. The GCD + JSON Grammar method uses only the JSON grammar (not specific to the schema), while the GCD + JSON Schema method uses the grammar generated from the JSON schema.

| Model | Method | pass@$k$ | | | | | |
|---|---|---|---|---|---|---|---|
| | | 1 | 3 | 5 | 10 | 20 | 50 |
| Llama 3.2 1B instruct-tuned | Standard | 55.6 | 53.9 | 60.0 | 67.1 | 73.5 | 80.8 |
| | GCD + JSON Grammar | 56.6 | 55.9 | 61.9 | 68.1 | 73.7 | 80.8 |
| | GCD + JSON Schema | **68.7** | **70.9** | **74.2** | **78.0** | **81.4** | **84.8** |
| Qwen2.5 0.5B instruct-tuned | Standard | 64.7 | 66.3 | 69.7 | 73.1 | 75.9 | 80.8 |
| | GCD + JSON Grammar | 64.7 | 67.0 | 70.8 | 74.8 | 77.8 | 81.8 |
| | GCD + JSON Schema | **67.7** | **69.8** | **73.0** | **77.2** | **81.2** | **85.9** |
| Gemma 3 270M instruct-tuned | Standard | 28.3 | 31.3 | 33.5 | 36.1 | 38.9 | 44.4 |
| | GCD + JSON Grammar | 29.3 | 33.2 | 35.6 | 38.6 | 41.4 | 45.5 |
| | GCD + JSON Schema | **56.6** | **62.1** | **64.2** | **66.2** | **67.6** | **68.7** |

### A.6.2 TEXT-TO-SQL GENERATION

We replicate the text-to-SQL generation task from Syncode (Ugare et al.).

**Dataset**  We use the Spider (Yu et al., 2018) dataset, which contains 1,034 text-to-SQL samples in the development set, the same as used in Syncode (Ugare et al.).

**Settings**  We compare two methods: unconstrained decoding (Standard) and grammar-constrained decoding (with PSC) using the SQL grammar (GCD).

**Models**  We use the same models as in the main experiments: Llama 3.2 1B, Qwen2.5 0.5B and Gemma 3 270M. We carefully construct the same prompt as in Syncode (Ugare et al.). Since the prompt does not contain chat-style instructions, we use the base versions of these models (not instruct-tuned).

**Metrics**  We use the standard execution accuracy (Exec Acc) (Zhong et al., 2020) metric for evaluation, the same as used in Syncode (Ugare et al.). We use greedy decoding to generate 1 SQL query for each sample, which is the default setting in Syncode (Ugare et al.).

**Grammar**  During the experiments, we found that the SQL grammar used in Syncode (Ugare et al.) cannot parse the `NOT` operator in the boolean expressions, making some valid SQL queries unparsable. We fixed this issue by adding the `NOT` operator to the grammar. The fixed grammar is provided in the open-sourced code.

**Results**  The results are presented in Table 6. For all the models tested, grammar-constrained decoding with PSC using the SQL grammar significantly outperforms unconstrained decoding (Standard). This confirms that grammar-constrained decoding can effectively improve the performance of LLMs on text-to-SQL generation tasks, and PSC can achieve this improvement while significantly speeding up the existing GCD methods.

### A.6.3 CODE GENERATION

We replicate the code generation task from Syncode (Ugare et al.).

**Dataset**  We use the Go subset of Multilingual HumanEval (Athiwaratkun et al.; Chen et al.) dataset, which contains 160 samples.

Table 6: The execution accuracy of different methods of text-to-SQL generation on the Spider dataset using grammar-constrained decoding (GCD) and unconstrained decoding (Standard). For the grammar-constrained decoding methods, we use PSC to compute the valid token masks.

| Model | Method | Execution accuracy | | | | |
| | | Easy | Medium | Hard | Extra Hard | Overall |
|---|---|---|---|---|---|---|
| Llama 3.2 1B | Standard | 35.9 | 25.3 | **14.9** | **6.6** | 23.1 |
| | GCD | **38.7** | **27.8** | **14.9** | **6.6** | **24.9** |
| Qwen2.5 0.5B | Standard | **31.0** | 24.0 | **13.2** | **6.6** | 21.1 |
| | GCD | **31.0** | **24.2** | **13.2** | **6.6** | **21.2** |
| Gemma 3 270M | Standard | 0.4 | 0.4 | 0.0 | 0.6 | 0.4 |
| | GCD | **1.6** | **1.3** | **0.6** | **1.8** | **1.4** |

**Settings**  We compare two methods: unconstrained decoding (Standard) and grammar-constrained decoding (with PSC) using the Go grammar (GCD).

**Models**  We use the same models as in the main experiments: Llama 3.2 1B, Qwen2.5 0.5B and Gemma 3 270M. We use the base versions of these models (not instruct-tuned), because this is a code completion task, and the prompt does not contain chat-style instructions.

**Metrics**  We use the standard pass@k metric (Chen et al.) for evaluation. For each sample, we generate 1 code completion with greedy decoding, and generate 50 completions with sampling (temperature 1.0, no top-$p$ or top-$k$ filtering). We report the pass@$k$ metric for $k = 1, 3, 5, 10, 20, 50$. For pass@1, we use the greedy decoding result; for pass@k where $k > 1$, we use the sampling results.

Table 7: The pass@$k$ scores (%) of different methods on Multilingual HumanEval Go dataset using grammar-constrained decoding (GCD) and unconstrained decoding (Standard). For the grammar-constrained decoding methods, we use PSC to compute the valid token masks.

| Model | Method | pass@$k$ | | | | | |
| | | 1 | 3 | 5 | 10 | 20 | 50 |
|---|---|---|---|---|---|---|---|
| Llama 3.2 1B | Standard | **5.6** | 4.0 | 5.4 | 7.4 | 9.3 | 11.9 |
| | GCD | **5.6** | **4.7** | **6.3** | **8.6** | **10.6** | **13.1** |
| Qwen2.5 0.5B | Standard | **8.8** | 5.4 | 7.2 | 9.9 | 12.4 | 15.6 |
| | GCD | **8.8** | **6.0** | **7.9** | **10.5** | **13.2** | **16.3** |
| Gemma 3 270M | Standard | **0.6** | 0.8 | 1.0 | 1.3 | 1.8 | 2.5 |
| | GCD | **0.6** | **1.0** | **1.3** | **1.9** | **2.9** | **5.0** |

**Results**  The results are presented in Table 7. For all the models tested, grammar-constrained decoding with PSC using the Go grammar significantly outperforms unconstrained decoding (Standard). This confirms that grammar-constrained decoding can effectively improve the performance of LLMs on code generation tasks, and PSC can achieve this improvement while significantly speeding up the existing GCD methods.

### A.7  EXTRA THROUGHPUT RESULTS

Due to page limit, we only present the end-to-end throughput results on Java and schema-conformant JSON in Section 4.3. The results on the Go and SQL datasets are similar and are included here in Table 4 and Table 5, respectively.

### A.8  PREPROCESSING OVERHEAD DETAILS

Whether the preprocessing overhead is acceptable depends on the specific application scenario.

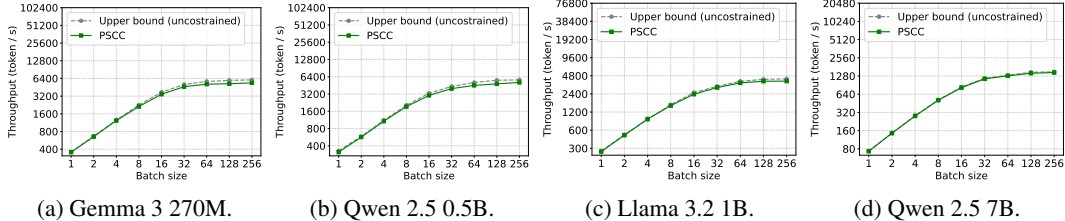

(a) Gemma 3 270M.  (b) Qwen 2.5 0.5B.  (c) Llama 3.2 1B.  (d) Qwen 2.5 7B.

Figure 4: End-to-end throughput (tokens per second) on the **Go** dataset using different methods on different models with various batch sizes.

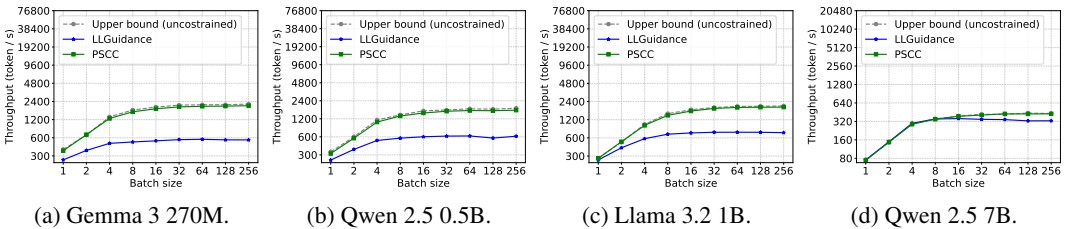

(a) Gemma 3 270M.  (b) Qwen 2.5 0.5B.  (c) Llama 3.2 1B.  (d) Qwen 2.5 7B.

Figure 5: End-to-end throughput (tokens per second) on the **SQL** dataset using different methods on different models with various batch sizes.

**Preprocessing time**   The preprocessing time can be amortized over multiple decoding sessions since it only needs to be done once per grammar and vocabulary pair. We calculate the time users need to use PSC for decoding to make it more time-efficient than using LLGuidance, our fastest baseline in the experiments. We reuse the throughput results from Section 4.3. It should be noted that this includes the total decoding time for all sequences, not just on one sequence.

In other words, we calculate the minimum $t_{\text{runtime}}$ such that,

$$t_{\text{runtime}} \cdot \text{throughput}_{\text{PSC}} \geq (t_{\text{preprocess}} + t_{\text{runtime}}) \cdot \text{throughput}_{\text{LLGuidance}},$$

where $t_{\text{preprocess}}$ is the preprocessing time of PSC on the grammar, and $\text{throughput}_{\text{PSC}}$ and $\text{throughput}_{\text{LLGuidance}}$ are the end-to-end throughput of PSC and LLGuidance, respectively. Rearranging the equation gives,

$$t_{\text{runtime}} \geq \frac{t_{\text{preprocess}} \cdot \text{throughput}_{\text{LLGuidance}}}{\text{throughput}_{\text{PSC}} - \text{throughput}_{\text{LLGuidance}}}.$$

The results are presented in Table 8. For Java and JSON schemas, the preprocessing time is generally small, and the balance point is within half to three minutes of decoding time. For SQL, the preprocessing time is higher, but the balance point is within fifty minutes of decoding time.

Thus, if the user plans to perform grammar-constrained decoding for a total time longer than the balance point, using PSC is more time-efficient than using LLGuidance. We believe that in many practical applications, users may perform grammar-constrained decoding for more than these balance points, making the preprocessing time acceptable.

**Memory usage**   The memory usage during preprocessing is generally higher than that during decoding. However, since the preprocessing can be performed offline, it does not affect the online decoding efficiency or memory usage. The preprocessing can be performed on cloud instances with large memory capacity if needed. For example, for SQL grammar preprocessing which requires around 250 GiB of memory, cloud providers like AWS offer instances with 500 GiB of memory for on-demand usage, and the cost is around 2.5 USD per hour. Since the preprocessing only needs to be done once per grammar and vocabulary pair, the cost is generally acceptable for practical applications.

Table 8: The time (in seconds) the user needs to use PSC to amortize the preprocessing time compared to using LLGuidance. Only results on Java, SQL and JSON schemas are shown here, because LLGuidance fails on Go as shown in Table 1.

| Model | Grammar | | |
|---|---|---|---|
| | Java | SQL | JSON Schemas |
| Llama 3 1B | 146.6 | 2826.2 | 25.1 |
| Qwen2.5 0.5B | 101.5 | 2760.7 | 20.1 |
| Gemma 3 270M | 67.2 | 508.5 | 40.0 |

