# OpenReview forum: "PSC: Efficient Grammar-Constrained Decoding via Parser Stack Classification"
_ICLR.cc/2026/Conference — Submitted to ICLR 2026_

### Official Review · Reviewer_J9vJ · 2025-10-24

**Soundness:** 3
**Presentation:** 2
**Contribution:** 3
**Rating:** 2
**Confidence:** 4

**Summary:**

This paper proposes a new approach for speeding up grammar constrained decoding.
The key insight is a clever one, instead of "running" the parser alongside with the decoder, one only needs to understand where in the parser the decoder in terms of stack state, and there are only finitely many "classes" of stack states that are interesting in terms of masking tokens away to perform constrained decoding. Furthermore, as known from theory of computation, these equivalence classes are recognizable by regular languages over stack traces.
One can therefore precompute all masking and do them very efficiently at decoding time.
The paper shows how this approach incredibly fast in practice (less than 3 microseconds overhead per token), thus defeating all the state of the art approaches.
The results only include speed of decoding, but not the so called "time-to-first-token" and is therefore not know what the pre-processing time of the tool.

**Strengths:**

- Clever idea of only considering regular languages of stack traces so one can avoid computing masks at runtime
- Evaluation shows clear gain over state of the art in terms of speed and in a way sets an unbeatable bar for future approaches.

**Weaknesses:**

The paper has two main problems, one fixable and one major.

Fixable problem:
The exposition of the paper is really really hard to follow. I knew the algorithm after the abstract (probably because I had thought about the same idea before, though I didn't expect it would have such a strong impact in time), and yet I couldn't follow the presentation.
Many symbols are undefined (e.g., I had to guess that \Pi is the set of stack symbols), the notion of "stabilized version" and stable stack are never defined, I don't know what {Final} means.
Many references to theorems and definitions are incorrect within the paper, making me guess where to read most of the time. Theorem 1 mentions Definition 4, but there are no definitions (so I assume it means equations). Similar issue with theorem 2. Line 274 refers to "simplification 3" but again, maybe it means equation and same for Definition 1 one line before. The algorithm are sort of given "as is" without an example of the intuition behind them. I would remove all the algorithms and put them in the appendix and focus the presentation on showing an example.

Major problem:
Now comes what I think is big limitation of the paper, it "hides" the cost of preprocessing. I'm 99% sure that this approach has very very high preprocessing time. Yet, I would be ok with it if it wasn't that the paper decided to hide this aspect and just say briefly in the conclusion "so the overhead of preprocessing is generally acceptable for practical applications". The tools the authors cite and compare against LLGuidance and GreatGramma ALL report time to first token (aka preprocessing) and time per token. So it is clear that people care about both of these quantities. In fact, GreatGrammar is explicitly built to balance the tradeoff between these two quantities. Without reporting preprocessing times, this paper CANNOT be accepted. On a related note, the authors should discuss the computational complexity of the approach.

There are some applications where preprocessing may not matter because the grammar is set once and for all at the beginning, but in most cases people use the grammar to "program" the output of the LLM and often do so at running time (e.g., they specialize the grammar to certain variable names or namespaces).

**Questions:**

What is the preprocessing time of PSC on all the grammars discussed in the paper and how does it compare to other tools?
(Once I see these numbers, I can form an opinion on whether I should update my score)

What is the computational complexity of the construction in PSC?

---

> ### Author Response · Authors · 2025-11-20
>
> We thank you for your review, and the relevant discussion will be included in the revised paper.
>
> # Weakness 1: Writing
>
> We do think our presentation is not good enough for the readers to understand. In the revised paper, we will rewrite it to make it easier to follow as much as we can.
>
> 1. The definition of the symbols and notions are placed in the Appendix due to page limit. We think this is a bad idea, and we will compress other parts of the paper to move the definitions into the main text of the paper in the revised paper.
> 2. We will try to add toy examples in paper to help understanding.
>
> # Weakness 2: Preprocessing
>
> We think offline preprocessing time is generally affordable. Preprocessing only needs to be done once on any machine, and the preprocessing result can be shared by different machines. We have updated our repository to provide the preprocessing results used in our experiments, and the user only needs to download the preprocessing results to use them.
>
> ## Preprocessing time
>
> |  |Java|Go|SQL|JSON schemas|
> |--|--|--|--|--|
> |Llama 3|466.9 s|1171.7 s|4662.7 s|28.3 s|
> |Qwen2.5|464.5 s|1166.8 s|4770.0 s|28.6 s|
> |Gemma 3|472.1 s|1362.2 s|1367.4 s|53.2 s|
>
> For all the programming languages, the preprocessing time of different models ranges from 8 minutes to 1.3 hours. In particular, only the preprocessing of SQL with Llama 3 and Qwen 2.5 takes an extremely long time of 1.3 hours. Since these grammars are almost fixed, preprocessing only needs to be done once. Although it is not so short compared to existing work, it is still affordable.
>
> For the JSON schemas, the average preprocessing time of different models ranges from half to one minute. Even if the JSON schemas might often need changes, the preprocessing time is still affordable.
>
> ### Trade-off between preprocessing cost and runtime cost
>
> Because, at runtime, our method runs significantly faster than LLGuidance (the best baseline in our experiments), the total time advantage of our method against LLGuidance becomes larger when we need to use the model for a longer time. We calculate the time needed for the model to run with PSC to make it more time-efficient than LLGuidance. In other words, we calculate $t_R$ in the following equation. Note that it means the **total** time the model is used (not only on one sequence).
>
> $$t_R\times T_{PSC}\geq(t_R+t_P)\times T_{LLGuidance},$$
>
> where $t_R$ means the time used in runtime, $t_P$ means the time used in preprocessing by PSC, $T_{PSC}$ means the throughput of PSC, and $T_{LLGuidance}$ means the throughput of LLGuidance.
>
> |  |Java|SQL|JSON schemas|
> |--|--|--|--|
> |Llama 3 1B|146.6 s|2826.2 s|25.1 s|
> |Qwen2.5 0.5B|101.5 s|2760.7 s|20.1 s|
> |Gemma 3 270M|67.2 s|508.5 s|40.0 s|
>
> Even for SQL where the preprocessing time is longest, if the user needs to run the model for more than one hour, using PSC is more time-efficient than LLGuidance.
>
> ## Preprocessing complexity
>
> We only prove that the preprocessing terminates. There is no bound on the computational complexity of PSC, but from the time given above, for JSON schemas (where grammar often changes), the time (less than one minute) is acceptable; for programming languages, the grammar is set once and never changed, so we think the time (8 minutes to 1.3 hour) is also acceptable.

---

> > ### Comment · Reviewer_J9vJ · 2025-11-27
> >
> > Thanks for the response. Overall I like the ideas in this paper, but it will require a major rewrite to incorporate the preprocessing and acknowledging the limitations.
> >
> > I don't agree with this sentence
> > "for programming languages, the grammar is set once and never changed, so we think the time (8 minutes to 1.3 hour) is also acceptable."
> > I can imagine many cases where the grammar is "generated" from the prompt to perhaps target different components in the code.

---

### Official Review · Reviewer_hYth · 2025-10-31

**Soundness:** 3
**Presentation:** 3
**Contribution:** 2
**Rating:** 4
**Confidence:** 4

**Summary:**

The paper proposes PSC, a GCD method that precomputes a finite-state automaton mapping parser stacks to valid tokens, reducing decoding time complexity.

**Strengths:**

The paper focuses on improving the efficiency of grammar-constrained decoding, and the writing is clear and easy to follow.

**Weaknesses:**

Novelty clarification: The core idea of PSC, precomputing finite-state representations of grammar validity, is well established in compiler theory and automata-based parsing. The paper would benefit from clarifying what is genuinely new about PSC beyond these known concepts. Also in practice, token masking is often not the main bottleneck compared to the model’s forward pass, so the real-world efficiency gains may be less pronounced.

Distributional bias and downstream performance: The paper focuses on decoding efficiency but does not examine whether enforcing strict grammatical validity alters the model’s output distribution and therefore degrades the downstream task performance. GCD is known to introduce such biases [1], so evaluating downstream quality, e.g., code accuracy, semantic correctness, or task performance, would make the results more complete.

Practical limitations: It would be helpful to discuss the preprocessing time and memory footprint of building and minimizing the automaton (reporting the total cost of the entire process rather than only the decoding phase), as well as how PSC handles dynamically changing grammars or schema updates. Quantifying these trade-offs would improve the paper’s transparency and practical relevance.

[1] https://arxiv.org/abs/2405.21047

**Questions:**

See above weakness.

---

> ### Author Response · Authors · 2025-11-20
>
> We thank you for your review. We will add the relevant discussion in the paper.
>
> ## Weakness 1: Novelty clarification
>
> **Novelty**: We think the main algorithmic contribution is our use of FSAs to model the stack condition for an DPDA to accept any terminal sequence. The combination of FSAs and masks are not possible without similar modeling of the stack condition for an DPDA to accept any terminal sequence. After the preprocessing, our method can achieve O(1) runtime efficiency for GCD.
>
> **Real-world efficiency gains**: The real-world efficiency gain depends on the throughput of the language model. When the throughput of the language model is high enough, the constrained decoding method can be the bottleneck. This is true when using a small model with a fast inference framework (e.g. vllm), and this is the case we present in Section 4.3.
>
> ## Weakness 2: Distributional bias and downstream performance
>
> **Distributional bias**: Grammar-Constrained Decoding is a practical feature in many generation frameworks, so we think it is generally important to improve its efficiency for these purposes.
>
> The core problem of distributional bias is not how the constraint mask is calculated or what constraint is used, but **how the mask is used in decoding**. The distributional bias happens when the mask is applied greedily.
>
> On the other hand, our method is useful **whenever a mask based on grammar is needed**. This includes **both** the traditional grammar-constrained decoding methods, and the new methods (e.g. Grammar-aligned decoding) that aim to reduce the distributional bias introduced by GCD. In other words, our method is orthogonal to research that aims at reducing the distributional bias, and can be combined with them.
>
> **Downstream performance**: The downstream performance of our method is the same as existing grammar-constrained decoding methods, because we only improve the runtime calculation efficiency, but our method provides the same masks. Nevertheless, we will provide the task improvements in the revised paper.
>
> ## Weakness 3: Practical limitations
>
> ### Preprocessing
>
> We think offline preprocessing time is generally affordable. Preprocessing only needs to be done once on any machine, and the preprocessing result can be shared by different machines. We have updated our repository to provide the preprocessing results used in our experiments, and the user only needs to download the preprocessing results to use them.
>
> #### Preprocessing time
>
> |  |Java|Go|SQL|JSON schemas|
> |--|--|--|--|--|
> |Llama 3|466.9 s|1171.7 s|4662.7 s|28.3 s|
> |Qwen2.5|464.5 s|1166.8 s|4770.0 s|28.6 s|
> |Gemma 3|472.1 s|1362.2 s|1367.4 s|53.2 s|
>
> For all the programming languages, the preprocessing time of different models ranges from 8 minutes to 1.3 hours. In particular, only the preprocessing of SQL with Llama 3 and Qwen 2.5 takes an extremely long time of 1.3 hours. Since these grammars are almost fixed, preprocessing only needs to be done once. Although it is not so short compared to existing work, it is still affordable.
>
> For the JSON schemas, the average preprocessing time of different models ranges from half to one minute. Even if the JSON schemas might often need changes, the preprocessing time is still affordable.
>
> #### Preprocessing memory footprint
>
> |  |Java|Go|SQL|JSON schemas
> |--|--|--|--|--|
> |Llama 3|40.8 GB|87.7 GB|255.3 GB|3.04 GB|
> |Qwen2.5|40.4 GB|86.6 GB|254.2 GB|3.13 GB|
> |Gemma 3|36.0 GB|88.8 GB|188.7 GB|5.95 GB|
>
> This is not optimized on purpose, because the main focus is the runtime overhead.
>
> The ~250GB is not always necessary, because the final FSA $\mathcal{A}$ in our implementation is constructed and minimized by parts. Splitting it into more parts reduces the preprocessing memory demands, but requires longer preprocessing time. A user with less memory can split it into more parts.
>
> The memory footprint for programming languages is large, but they do not need to be done on the runtime machine. Using another server for preprocessing is also possible. For example, using an Amazon EC2 r6g.12xlarge instance with 384 GB memory for 2 hours, one can handle preprocessing for \$4.84.
>
> ### Grammar changes or updates
>
> When the grammar changes or updates, preprocessing needs to be done again. Grammar changes or updates do not happen often for programming languages, so the preprocessing overhead should be affordable. As for the changes/updates of JSON schemas, the overall perprocessing overhead is small enough for frequent changing.
>
> In the future, we will explore how to reuse existing preprocessing results for grammar updates or changes.

---

### Official Review · Reviewer_uP4S · 2025-11-01

**Soundness:** 2
**Presentation:** 1
**Contribution:** 2
**Rating:** 2
**Confidence:** 4

**Summary:**

The paper aims to reduce the processing time of grammar-constrained decoding (GCD). The authors argue that existing approaches must check every token in the vocabulary against the parser to ensure syntactic validity, leading to an O(|V|) runtime complexity—though this claim is somewhat overstated, as many prior methods already exploit token tries or sparse validity masks. The proposed method, Parser Stack Classification (PSC), precomputes the results of the lexing and parsing process for all possible combinations of parser states and tokens (or terminal sequences). At runtime, PSC avoids re-running the parser and instead retrieves precomputed validity results from a large lookup structure, achieving up to 770× speedup in mask computation and significant throughput gains.

The core contribution is not conceptual novelty but rather a proof-of-extremes: the paper demonstrates that, if one is willing to precompute and store enough information, the runtime cost of grammar checking can be minimized almost completely

**Strengths:**

The paper tackles a practically relevant problem — reducing the runtime cost of grammar-constrained decoding. The authors demonstrate clear effort in building a formal framework (although overstretched and overcomplicated) that connects the lexer, parser, and grammar validity checking. The underlying idea of trading runtime complexity for offline precomputation is conceptually reasonable and aligns with longstanding efficiency strategies in compiler and parsing theory. However, while the overall idea is sound the approach leans too heavily on exhaustive precomputation to claim constant-time decoding. Still, the paper's motivation, completeness of implementation, and inclusion of multiple baselines reflect solid execution effort.

**Weaknesses:**

1. Computation is not eliminated, only shifted to preprocessing.
The paper's central claim—that PSC achieves "runtime independent of |V|"—is misleading. The method still performs preprocessing for every token and ultimately stores a full |V|-sized mask for each combined lexer–parser state. In effect, it trades O(|V|) runtime cost for O(|V|) offline computation and storage. This does not save computation—it merely moves it earlier in the pipeline. Moreover, many of the precomputed state transitions will never occur at runtime, making the approach less efficient overall. This is conceptually similar to precomputing all n×n matrix multiplications in order to claim O(1) inference—a theoretically valid but practically meaningless optimization.
The paper also provides no quantitative discussion of precomputation time or memory footprint, despite these being the dominant costs. Given that the automaton A stores a |V|-bit mask per stack-equivalence class, the storage requirements for nontrivial grammars can explode rapidly. Without measurements of preprocessing time, number of automaton states, or disk/memory usage, the practicality of PSC remains unsubstantiated.

Finally, the authors' criticism of prior work ("existing methods compute over all tokens") is not accurate. Many grammar-constrained decoding systems already use trie-based lexers or incremental constraint propagation, achieving complexity proportional to the number of valid tokens rather than the full vocabulary. PSC's precomputation does not strictly improve upon these optimizations and may in fact be less efficient when constraints are sparse.

⸻

2. Low novelty relative to existing approaches.
Several prior GCD systems (e.g., XGrammar, GreatGramma, Formatron) already perform offline caching of terminal sequences, context-independent masks, or parser states. The claim that "previous approaches cannot fundamentally change O(|V|)" is true yet not really meaningful, given that many already reduce effective runtime complexity through structured token tries or hierarchical lexers. Overall, the idea of moving computation all to preprocessing is limited in novelty and not creative.

⸻

3. Overcomplicated and low-yield theoretical presentation.
Part of the paper's exposition is unnecessarily formal. Sections 3.3–3.5 introduce multiple "theorems" that are self-evident ("by construction") and add little technical insight, while essential notation (e.g., Π for the stack alphabet) is deferred to the appendix. The presentation is cluttered with symbolic machinery—ε-FSTs, multi-level DFAs, nested composition operators—that obscure a relatively simple idea: precompute parser transitions and look up validity masks at runtime. Compared with prior work such as XGrammar, the writing here is substantially less readable and less disciplined in its abstraction boundaries. Simplifying the derivations and focusing on empirical scalability would make the contribution far clearer and more impactful.

——

While the paper's motivation—achieving faster grammar-constrained decoding—is clear, the work is fundamentally limited by its blind pursuit of "runtime efficiency." PSC achieves constant-time decoding only by offloading essentially all computation into an enormous precomputation phase. This tradeoff is not just theoretically questionable but also practically unappealing: it replaces a manageable online cost with unbounded preprocessing time and memory demands. Although such precomputation is technically possible, it is neither interesting from a research perspective nor useful in real deployments. The authors should provide a much deeper discussion on the balance between runtime efficiency and pre-runtime cost, including when such a tradeoff is actually worthwhile and how the claimed runtime benefit compares quantitatively to the enormous precomputation overhead. A convincing evaluation would require detailed reporting of:

- **Wall-clock precomputation time:** how long it takes to build P_\varepsilon, \tilde P_a, all P_w, and the final automaton A.
- **Automaton size:** number of states |A| before and after minimization.
- **Disk or RAM usage:** total bytes of transition tables or per-state vocabulary masks.
- **Scaling curves:** how preprocessing scales with grammar complexity or vocabulary size (|Γ|, |Π|, |V|).
- **Compression results:** any mention of FSA minimization effectiveness or disk savings.

And the above is certainly grammar dependent so the reporting should also span a variety of grammar in terms of the number of parser and stack states.

**Questions:**

See weaknesses

---

> ### Author Response · Authors · 2025-11-20
> **Preprocessing overhead**
>
> We thank you for your review. All the statistics and discussion will be included in the revised paper. Our comment is split into several comments due to length limits.
>
> ## Preprocessing overhead
>
> We think offline preprocessing time is generally affordable. Preprocessing only needs to be done once on any machine, and the preprocessing result can be shared by different machines. We have updated our repository to provide the preprocessing results used in our experiments, and the user only needs to download the preprocessing results to use them.
>
> ### Preprocessing time
>
> |  |Java|Go|SQL|JSON schemas|
> |--|--|--|--|--|
> |Llama 3|466.9 s|1171.7 s|4662.7 s|28.3 s|
> |Qwen2.5|464.5 s|1166.8 s|4770.0 s|28.6 s|
> |Gemma 3|472.1 s|1362.2 s|1367.4 s|53.2 s|
>
> For all the programming languages, the preprocessing time of different models ranges from 8 minutes to 1.3 hours. In particular, only the preprocessing of SQL with Llama 3 and Qwen 2.5 takes an extremely long time of 1.3 hours. Since these grammars are almost fixed, preprocessing only needs to be done once. Although it is not so short compared to existing work, it is still affordable.
>
> For the JSON schemas, the average preprocessing time of different models ranges from half to one minute. Even if the JSON schemas might often need changes, the preprocessing time is still affordable.
>
> #### Trade-off between preprocessing cost and runtime cost
>
> Because, at runtime, our method runs significantly faster than LLGuidance (the best baseline in our experiments), the total time advantage of our method against LLGuidance becomes larger when we need to use the model for a longer time. We calculate the time needed for the model to run with PSC to make it more time-efficient than LLGuidance. In other words, we calculate $t_R$ in the following equation. Note that it means the **total** time the model is used (not only on one sequence).
>
> $$t_R\times T_{PSC}\geq(t_R+t_P)\times T_{LLGuidance},$$
>
> where $t_R$ means the time used in runtime, $t_P$ means the time used in preprocessing by PSC, $T_{PSC}$ means the throughput of PSC, and $T_{LLGuidance}$ means the throughput of LLGuidance.
>
> |  |Java|SQL|JSON schemas|
> |--|--|--|--|
> |Llama 3 1B|146.6 s|2826.2 s|25.1 s|
> |Qwen2.5 0.5B|101.5 s|2760.7 s|20.1 s|
> |Gemma 3 270M|67.2 s|508.5 s|40.0 s|
>
> Even for SQL where the preprocessing time is longest, if the user needs to run the model for more than one hour, using PSC is more time-efficient than LLGuidance.
>
> ### Preprocessing memory footprint
>
> |  |Java|Go|SQL|JSON schemas
> |--|--|--|--|--|
> |Llama 3|40.8 GB|87.7 GB|255.3 GB|3.04 GB|
> |Qwen2.5|40.4 GB|86.6 GB|254.2 GB|3.13 GB|
> |Gemma 3|36.0 GB|88.8 GB|188.7 GB|5.95 GB|
>
> This is not optimized on purpose, because the main focus is the runtime overhead.
>
> The ~250GB is not always necessary, because the final FSA $\mathcal{A}$ in our implementation is constructed and minimized by parts. Splitting it into more parts reduces the preprocessing memory demands, but requires longer preprocessing time. A user with less memory can split it into more parts.
>
> The memory footprint for programming languages is large, but they do not need to be done on the runtime machine. Using another server for preprocessing is also possible. For example, using an Amazon EC2 r6g.12xlarge instance with 384 GB memory for 2 hours, one can handle preprocessing for \$4.84.
>
> ### Disk usage of all the components
>
> We store the preprocessing results with zstandard.
>
> |  |Java|Go|SQL|JSON schemas|
> |--|--|--|--|--|
> |Llama 3|13.27 MB|28.16 MB|58.95 MB|549.51 KB|
> |Qwen2.5|12.70 MB|28.37 MB|57.60 MB|526.38 KB|
> |Gemma 3|13.13 MB|33.52 MB|24.04 MB|774.87 KB|
>
> The disk usage is in tens of MB for programming languages, and hundreds of KBs for JSON schemas, both affordable.
>
> ### Runtime memory usage of final DFAs $\mathcal{A}$
>
> |  |Java|Go|SQL|JSON schemas|
> |--|--|--|--|--|
> |Llama 3|162.8 MB|752.2 MB|3.42 GB|296.8 KB|
> |Qwen2.5|165.2 MB|763.4 MB|3.37 GB|284.6 KB|
> |Gemma 3|124.3 MB|932.9 MB|1020.4 MB|288.7 KB|
>
> Here is the runtime memory usage of precomputed masks.
>
> |  |Java|Go|SQL|JSON schemas|
> |--|--|--|--|--|
> |Llama 3|227.3 MB|1.68 GB|2.82 GB|1.84 MB|
> |Qwen2.5|264.5 MB|1.98 GB|3.33 GB|2.02 MB|
> |Gemma 3|457.4 MB|6.32 GB|2.78 GB|3.31 MB|
>
> We think the sizes of these DFAs and masks are affordable for a modern system.
>
> ## How the preprocessing cost scales with the |Γ|, |Π|, |V|
>
> The number of lexer states and parser states are provided below.
>
> |  |Java|Go|SQL|JSON schemas|
> |--|--|--|--|--|
> |Lexer states|292|211|328|111.85|
> |Parser states|660|881|764|275.57|
>
> The exact preprocessing cost is not simply determined by |Γ|, |Π|, |V| alone, but is determined by the internal structure of the grammar and the vocabulary. For example, the preprocessing time for Go and SQL is close for Gemma 3, but very different for Llama 3 and Qwen 2.5. On the other hand, the preprocessing time for Java is close for all 3 tokenizers, but the preprocessing time for SQL is very different for Gemma 3 and the other 2 models.

---

> > ### Comment · Reviewer_uP4S · 2025-11-27
> >
> > Thanks to the authors for the detailed response and for providing the additional experimental results.
> >
> > I agree that the preprocessing time is acceptable for both programming languages and JSON schemas. Trading preprocessing time for runtime efficiency is a reasonable design choice, and the introduction of t_R is a helpful and intuitive metric.
> >
> > Regarding memory footprint, although it is relatively large, I think it is still practical, especially given that preprocessing can be done on a separate server. The runtime memory usage is also sufficiently small and does not raise concerns.
> >
> > I appreciate the additional effort the authors have put into address my concern and I am raising my score from 2 to 4. I would also like to lower my confidence from 4 to 3, as I now have less certainty in my evaluation. My main reason for not raising to 6 is that I still think the write-up is not easy to follow and should be improved for clarity.

---

> ### Author Response · Authors · 2025-11-20
>
> Thank you for your review. This is the second part of our comment.
>
> ## Computation
>
> "Computation is [...] shifted to preprocessing" is true, but the computation has to be done somewhere at some time. To avoid the runtime overhead, the option we provide is to move it to preprocessing. Many efforts are put into this direction, including the context-independent tokens in XGrammar or context-independent terminal sequences in GreatGramma, which are all identified during preprocessing. We follow the same direction. By using the FSAs to model the requirements on the DPDA stack, we provide the "extreme" in this direction.
>
> ## Related work
>
> We did not say that "existing methods compute over all tokens", but we do say that their runtime complexity is O(|V|), and we have summarized the optimizations of related work in Section 2.
> In particular, trie-based lexers are summarized as "Vocabulary preprocessing".
> Although trie-based lexers can skip tokens whose prefix is rejected, it still has to calculate the constraint for **the prefix** at runtime. On the other hand, our method does not need such calculation at all at runtime, which leads to theoretically better runtime performance. In JSON schemas, most field names are constant, which makes it similar to a sparse constraint. But our method beats LLGuidance in JSON schemas in the experiments, showing the same consequence experimentally.
>
> ## Weakness 2: Novelty
>
> As we have stated, many efforts are put into reducing runtime overhead. By saying "they cannot fundamentally change O(|V|)", **we do not mean their contributions are ignorable**. Rather, our method are in **the same direction** of exploring context-independent tokens, similar to XGrammar and GreatGramma, and takes one step further to completely get rid of context-dependent tokens.
>
> We think the main algorithmic contribution is our use of FSAs to model the stack condition for an DPDA to accept any terminal sequence. The combination of FSAs and masks are not possible without a similar modeling of the stack condition for an DPDA to accept any terminal sequence. XGrammar and GreatGramma try to explore context-independent tokens, but they do not move all computation into preprocessing, because their "context-dependency" is only based on the stack top state. By the novel idea of using FSAs to model the "context", we successfully move all computation into preprocessing, and get rid of most of the runtime overhead. This should be considered an algorithmic contribution of our work.
>
> ## Weakness 3: Presentation
>
> We do think our presentation is not good enough for the readers to understand. The definition of the symbols and notions are placed in the Appendix due to page limit, and the theoretical presentation might seem boring, trivial, or difficult to follow. We will compress other parts of the paper to move the definitions into the main text of the paper, and make the presentation easier to follow as much as we can in the revised paper.
>
> On the other hand, we do think **a** theoretical presentation is necessary (but not necessarily the same as what we present in the paper). We use FSTs to model the state transitions of DPDA, but this is not trivial, and not self-evidently correct by simply saying "precompute parser transitions". The question is "why and how one can use FSTs to model the state transitions", which is provided by the theoretical presentation.

---

### Official Review · Reviewer_8qp2 · 2025-11-02

**Soundness:** 3
**Presentation:** 3
**Contribution:** 3
**Rating:** 6
**Confidence:** 5

**Summary:**

The paper presents an efficient grammar-constrained LLM generation technique called PSC. Given any partial generation PSC computes the set of acceptable tokens according to the grammar in following steps. For each token, PSC precomputes a FSA that represents the condition for parser state to accept the token. These FSAs are combined into single FSA that maps parser state to vocabulary masks.

During decoding, a lexer is used to compute lexical tokens corresponding to the partial generation. This is used to compute the parser state. Using the encoded mapping through the FSA, a vocabulary mask is obtained efficiently by a simple lookup. The evaluation results show that - while small language models constraining on grammars for JSON, Java, Go, SQL generation PSC outperforms prior state-of-the-art techniques such as Xgrammar, Formatron, GreatGramma and LLGuidance in terms of efficiency.

**Strengths:**

1) There have been several grammar-constrained generation works in recent years. The grammar-constrained generation techniques have been commonly used for function-calling and it has become standard to support this feature on all LLM serving engines. PSCs throughput results are impressive and would likely help in further adoption of constrained-generation techniques for other more complex programming language grammars.

2) The presentation is detailed and the guarantees provided by the technique are proved rigorously.

3) The throughput experiments consider state-of-the-art baselines such as Xgrammar, Formatron, GreatGramma and LLGuidance and shows improvement over them.

Overall, I’m positive about the paper and happy to increase my score further if the authors could address my remaining concerns.

**Weaknesses:**

1) The paper does not report: (1) offline precomputation time for each grammar, (2) memory footprint of the combined FSAs, (3) how these scale with grammar complexity


2) Some of the claims about all prior works requiring $O(|V|)$ parsing steps in the worst case are inaccurate.

> However, none of these techniques can fundamentally change the time complexity, which is still $O(|V|)$ in the worst case.

Syncode (Ugare et. al.) performs single parsing step per decoding step as well. Syncode also computes set of acceptable terminal sequences (using LR parser) referred as accept sequences which is similar to realizable terminal sequences in PSC.

The main difference appears to be that Syncode requires mask lookup for each terminal sequence and a union operation over those masks, while PSC combines FSAs offline leading to a single lookup during inference. Can the authors confirm this characterization is accurate?

3) While PSC maintains syntactic correctness guarantees (Table 2), it's unclear whether the efficiency gains allow for practical improvements in downstream tasks.  Even if PSC does not lead to improved semantic correctness, I would encourage the authors to include an experiment for computing pass@k accuracy on standard code generation tasks.

Nits:

* Section 4.2 should mention the size of models used for the experiment
* I spotted a few places with missing whitespace after punctuations. Line 149, 482.

> Line 194: While some methods employ precomputation to optimize certain cases, they still fundamentally require

Precise citations here will be helpful.

**Questions:**

q1) Is the combination of FSAs the primary novel algorithmic contribution compared to GreatGemma and Syncode? Can you provide an ablation study focusing on efficiency gains from: algorithmic improvements such as FSA combination to disregard the gains from implementation optimizations like use Cython?

q2) What is the effect of lookahead of LALR parser in overall strength of PSC? Are there any downsides in using LALR parser instead of an Earley parser?

q3) What is the offline time taken for pre-computation performed by PSC for grammars considered in the evaluation?

q4) What is the memory footprint of the combined FSA structures for each grammar?

---

> ### Author Response · Authors · 2025-11-20
> **Response**
>
> We thank you for your careful review. We will fix all the typos. All the extra statistics and discussion will be included in the revised paper.
>
> # Weakness 1: Extra statistics
>
> ## Offline preprocessing time
>
> |  |Java|Go|SQL|JSON schemas|
> |--|--|--|--|--|
> |Llama 3|466.9 s|1171.7 s|4662.7 s|28.3 s|
> |Qwen2.5|464.5 s|1166.8 s|4770.0 s|28.6 s|
> |Gemma 3|472.1 s|1362.2 s|1367.4 s|53.2 s|
>
> The offline preprocessing time is generally affordable. Preprocessing only needs to be done once on any machine, and the result can be shared by different machines. We have provided the preprocessing results in our repository, and the user only needs to download the preprocessing results for their own purposes.
>
> For all the programming languages, the preprocessing time of different models ranges from 8 minutes to 1.3 hours. In particular, only the preprocessing of SQL with Llama 3 and Qwen 2.5 takes an extremely long time of 1.3 hours. Since these grammars are almost fixed, preprocessing only needs to be done once. Although the preprocessing time is not so short compared to existing work, it is still affordable.
>
> For the JSON schemas, the average preprocessing time of different models ranges from half to one minute. Even if the JSON schemas might often need changes, the preprocessing time is still affordable.
>
> ## runtime memory footprint of the final FSAs $\mathcal{A}$
>
> |  |Java|Go|SQL|JSON schemas|
> |--|--|--|--|--|
> |Llama 3|162.8 MB|752.2 MB|3.42 GB|296.8 KB|
> |Qwen2.5|165.2 MB|763.4 MB|3.37 GB|284.6 KB|
> |Gemma 3|124.3 MB|932.9 MB|1020.4 MB|288.7 KB|
>
> We think this is affordable for modern systems.
>
> ## How these scale with grammar complexity
>
> For the relatively complex programming languages, the offline preprocessing time and runtime memory footprint are large, but still affordable, since they are only preprocessed once and almost fixed afterwards.
>
> For the relatively simple JSON schemas, the offline preprocessing time and runtime memory footprint is small enough for frequent changing.
>
> # Weakness 2: Comparison with Syncode
>
> The runtime compelxity of Syncode is $O(|A|)$, where $|A|$ is the set of accept sequences.
>
> However, (1) this is achieved by only considering the accept sequences of length 1 and 2 using the lookaheads of LR(1) parser. Due to this approximation, Syncode can fail to create accurate masks, as found in [GreatGramma](https://openreview.net/forum?id=L6CYAzpO1k). (2) Syncode considers all the accept sequences of length 1 and 2, while most of them are not realizable, introducing unnecessary calculations. (3) The set of "accept sequences" in Syncode is large when all the realizable terminal sequences are considered, so the union operation can be expensive in this case.
>
> On the other hand, PSC can create accurate masks from LR(1) parsers, and only considering the realizable terminal sequences.
>
> # Weakness 3: Extra experiments on standard code generation tasks
>
> The downstream performance of our method will be the same as previous grammar-constrained decoding methods. Our method improves the efficiency of mask calculation, but we are calculating the same mask as the previous grammar-constrained decoding methods.
>
> Nevertheless, we thank your advice, and we will include the performance of PSC on downstream tasks in the revised paper.
>
> # Nits
> 1. Section 4.2: These experiments only measure the time used to calculate the mask, which only depends on the tokenizer (same for the same model series), so we only mention the name of the model series.
>
> # Questions 1: Algorithmic contribution
>
> We think the main algorithmic contribution is our use of FSAs to model the stack condition for an DPDA to accept any terminal sequence. The combination of FSAs and masks are not possible without similar modeling of the stack condition for an DPDA to accept any terminal sequence. After this, our preprocessing results can be used for generation at runtime.
>
> Here is the runtime latency with pure Python, close to the statistics reported in the paper using Cython.
>
> |  |Java|Go|SQL|JSON schemas|
> |--|--|--|--|--|
> |Llama 3|3.35 us|3.39 us|3.51 us|2.94 us|
> |Qwen2.5|3.33 us|3.41 us|3.36 us|2.84 us|
> |Gemma 3|3.22 us|3.27 us|3.36 us|2.83 us|
>
> # Question 2: LALR & Early
> In terms of expressiveness, Earley = PDA > DPDA = LR(1) = LR(n) > LALR(n). Our method is applicable to any LR parser. We use the LALR parser because Lark (our parser library) does not provide a Canonical LR parser. The "LookAhead" in LALR is compared to LR(0) parser, but LALR parser itself should be seen as a simplified version of LR parser.
>
> The language corresponding to DPDA is Determinstic Context-Free Grammar, which is a subset of Context-Free Grammars. Because of this, we cannot handle CFG that is not deterministic. (DCFG is sufficient for most programming languages.) Earley parser can handle all Context-Free Grammars, but cannot be viewed as a DPDA.
>
> In future, we will explore the integration of PSC and Earley parser.
>
> # Question 3 & 4
>
> They are answered in Weakness 1.

---

### Author Response · Authors · 2025-12-02

We have revised our paper to integrate all the parts that we promised to improve in the responses. The changed parts are marked blue to make it easier to identify.

## Preprocessing (all the reviewers)

We added a new section, Section 5, to talk about the preprocessing overhead of our method. We use the statistics and the logics that we have used in the response. We also added a new appendix, Appendix A.8, to discuss the balance between the preprocessing overhead and runtime overhead. As we have said in the responses, if one has to use the model for more than one hour, we think our method is more time-efficient even considering the preprocessing overhead.

On the other hand, we would address the concern of Reviewer J9vJ about "many cases where the grammar is "generated" from the prompt to perhaps target different components in the code". We acknowledge that such cases exist, but we do **not** think that our method needs to be the best choice in **every** situation.

**Our method is useful for all the use cases that we have shown in paper, which are also the use cases targeted by previous research**. Existing approaches, e.g. Syncode and GreatGramma, also need preprocessing that is not quick enough for the use cases pointed out by Reviewer J9vJ. In this sense, we would say that the preprocessing overhead of our method is acceptable for the use cases considered in previous research.

## Writing and the main method (reviewers uP4S and J9vJ)

1. We add the symbols and definitions into the new sections 2.2 and 2.3 to make the symbols and terms defined before being used. We also fix all the misused terms referencing equations to just equations.
2. We move algorithms and proofs about $\varepsilon$-FST and FSTs of terminal sequences into the appendix, and we only leave information about the general principle of building them in the main text. We believe it is more friendly to the readers now. On the other hand, although the Reviewer J9vJ recommends to add an example to ease understanding, we actually found it really hard to make an easy-to-understand example without using a ton of abstract states, so we decided to not add such an example.

## Downstream performance (reviewers 8qp2 and hYth)

We added a new appendix, Appendix A.6, to show the usefulness of grammar-constrained decoding. We replicated the experiments in Syncode, and we found that for all the experiments, the performance of grammar-constrained decoding is consistently better than unconstrained decoding. It should be noted that our method generates the same mask as the previous GCD engines, but only generates the mask faster. The performance increase should be considered a common result of GCD.

We would again clarify the concern raised by Reviewer hYth that "enforcing strict grammatical validity alters the model's output distribution and therefore degrades the downstream task performance". This is not true, as shown by our experiments. Distribution distortion introduced by grammar-constrained decoding does not degreade the downstream performance, but it just makes the result not as good as it **could** be, but still **better** than unconstrained decoding, and grammar-aligned decoding is about recovering the performance that it could be. Grammar-constrained decoding just removes the output that violates the grammar, and any output that violates the grammar should not be considered correct by definition.

## Syncode (Reviewer 8qp2)

We reconsidered the relationship between our method and Syncode. We read the its paper again, and add a short description about it in the section of related work.

We should say that, Syncode also precomputes the possible masks for each terminal sequence, and eliminates the dynamic parsing overhead by precomputing the accept sequences from the LR(1) parser lookaheads. However, Syncode needs an LR(k) parser for accept sequences of length k, making it unsuitable for tokens that might expand multiple terminals.

Syncode solves this problem by only considering the first two terminals of each token, so its mask might allow illegal tokens, as found in GreatGramma. GreatGramma solves this problem by only considering the realizable terminal sequences, but again introduces the need for dynamic parsing of O(|V|).

We solve this problem by modeling the accepting conditions of terminal sequences as FSAs, eliminating both the need for dynamic parsing and the construction of the LR(k) parsers.

---

### Meta-Review · Area_Chair_fUMe · 2026-01-07

**Summary:**

The paper shifts much of the cost of grammar constrained decoding from the runtime to preprocessing. The reviewers are mostly unconvinced, raising the following important weaknesses not addressed by the authors. The main problem is that this paper is optimizing something that may not be important enough to be optimized this way, so the trade offs and additional complexity introduced may be detrimental. The run-time scaling with vocabulary size in grammar checking is optimized, but there are important questions on if this was a problem to begin with before we go ahead and optimize it by applying existing knowledge.

> Also in practice, token masking is often not the main bottleneck compared to the model’s forward pass, so the real-world efficiency gains may be less pronounced.

This quite plausible point is not addressed.

> Distributional bias and downstream performance: The paper focuses on decoding efficiency but does not examine whether enforcing strict grammatical validity alters the model’s output distribution and therefore degrades the downstream task performance. GCD is known to introduce such biases [1], so evaluating downstream quality, e.g., code accuracy, semantic correctness, or task performance, would make the results more complete.

This point on the trade offs of doing such work is not addressed

> Major problem: Now comes what I think is big limitation of the paper, it "hides" the cost of preprocessing. I'm 99% sure that this approach has very very high preprocessing time...

The authors provides the preprocessing time table, but the reviewer disagrees that this really can only be done once. I think the reviewer has a point here, while particular programming language have a mostly fixed grammar. The most productive checks can indeed change.

**Reviewer Concerns:**

postprocessing is not sufficiently addressed

**Reviewer Scores:**

they participated. one reviewer increased from 2-4

---

### Decision · Program_Chairs · 2026-01-26

Reject